# Distinct magneto-optical response of Frenkel and Wannier excitons in CrSBr

**Maciej Śmiertka** [1], **Michał Rygała** [2], **Katarzyna Posmyk** [1,2], **Paulina Peksa** [1,2], **Mateusz Dyksik** [1], **Dimitar Pashov** [3], **Kseniia Mosina** [4], **Zdeněk Sofer** [4], **Mark van Schilfgaarde** [5], **Florian Dirnberger** [6,7,8], **Michał Baranowski** [1] ✉, **Swagata Acharya** [5] ✉ & **Paulina Plochocka** [1,2] ✉

Excitons in recently discovered two-dimensional magnetic semiconductors have emerged as a promising vehicle for optoelectronic and spin-photonic applications. To exploit novel possibilities magnetic degrees of freedom offer, insight into the interplay of magnetism, lattice and optical excitations becomes essential. We consider Chromium Sulphur Bromide, which has two kinds of excitons, $X_B$ at 1.8 eV and $X_A$ at 1.38 eV. Here we show, through a combination of many body perturbation theory and experiment, that $X_B$ is an order of magnitude more sensitive to magnetic and lattice perturbations than $X_A$. We trace the difference to the latter being localised (Frenkel-like), while the former is delocalised (Wannier-Mott-like) – a coexistence rarely seen in two-dimensional materials. This finding is supported by the strong temperature and magnetic field (up to 85 Tesla) dependent shifts in optical response for $X_B$ (much smaller for $X_A$), and we show it is related to $X_B$'s tendency for delocalisation (in-plane and out-of-plane) and enhanced coupling with A$g$ phonon modes.

Excitons, the excitation of an electron-hole pair bound by the screened Coulomb interaction, constitute a fundamental excitation in semiconductors. As they are the dominant optical response for technological applications in 2D semiconductors[1], understanding the structure of excitons and how they respond to external fields becomes an issue of central importance. 2D materials create fascinating playgrounds that continuously challenge our understanding of excitons, bringing into play such factors as nonuniform dielectric screening[2,3], enhanced exciton fine structure[4–6], or the formation of dipolar and hybridized excitons in van der Waals heterostructures as well as many-body interactions in moiré structures[7–9].

The recent discovery of magnetic order in atomically thin van der Waals (VdW) semiconductors[10–16] brings new degrees of freedom into 2D excitonic physics. Like in nonmagnetic materials, bound electron and hole pairs drive the optical response. Crucially, the excitonic states are coupled to the underlying magnetic state of the system; thus, the role of magnetic order in the optical response becomes of fundamental interest[13,17–22]. These magnetic excitons enhance spin-related phenomena, for instance, the Kerr effect[17,22], where their spectral signature can be directly related to the magnetic state of the material[18–20,23]. In CrSBr[24,25], a newcomer to the magnetic 2D family and the subject of this work, magnetic excitons are especially pronounced. The optical

[1]Department of Experimental Physics, Faculty of Fundamental Problems of Technology, Wroclaw University of Science and Technology, Wroclaw, Poland. [2]Laboratoire National des Champs Magnétiques Intenses, EMFL, CNRS UPR 3228, Université Grenoble Alpes, Université Toulouse, Université Toulouse 3, INSA-T, Grenoble and Toulouse, France. [3]King's College London, Theory and Simulation of Condensed Matter, The Strand, London, UK. [4]Department of Inorganic Chemistry, University of Chemistry and Technology Prague, Prague 6, Czech Republic. [5]National Laboratory of the Rockies, Golden, CO, USA. [6]Physics Department, TUM School of Natural Sciences, Technical University of Munich, Munich, Germany. [7]Zentrum für QuantumEngineering (ZQE), Technical University of Munich, Garching, Germany. [8]Munich Center for Quantum Science and Technology (MCQST), Technical University of Munich, Garching, Germany. ✉e-mail: michal.baranowski@pwr.edu.pl; swagata.acharya@nrel.gov; paulina.plochocka@lncmi.cnrs.fr

response of this layered, semiconducting A-type antiferromagnet is dominated by the intralayer confined excitons[26] whose energies track magnetic moment alignment in neighbouring layers[19,20,27–29].

Most classic and 2D semiconductors host delocalised excitons of the Wannier-Mott type[30], while excitons in magnetic insulators are typically strongly localised charge-transfer[31,32] or quasi-atomic Frenkel exciton[33–36]. They usually derive from a transition metal $d$ orbitals[17,18,23,31], which also carry the magnetic moment. Here, we demonstrate that CrSBr bridges the two distinct pictures of Wannier-Mott and *quasi*-Frenkel excitons, providing a new platform to investigate the interplay between different excitonic regimes and the resulting impact of magnetic order on optical response. Our study reveals the coexistence of Frenkel-like and Wannier-Mott-like excitons in CrSBr, and shows that their responses to magnetic and lattice perturbations vary dramatically. By employing high magnetic fields (up to B=85 T) in conjunction with state-of-the-art electronic structure calculations within the quasiparticle self-consistent $GW$ approximation augmented with ladder diagrams ($QSG\widehat{W}$)[37], we map out the spatial extensions of these two exciton types. We provide a microscopic understanding of the magneto-optical response of these distinct states and show that the Wannier-Mott-like exciton is ten times more sensitive as a probe of the magnetic order than the Frenkel-like exciton, a direct consequence of their delocalised character. Temperature-dependent studies reveal the critical role of the nature of excitons for coupling to lattice vibrations, which drives the temperature evolution of their optical response in both antiferromagnetic (AFM) and ferromagnetic (FM) phases. Importantly, our combined experimental and theoretical study highlights the limitations of conventional phenomenological models–such as molecular ligand-field theory[38,39] and Rydberg series[40,41]–in predicting the complex interplay between excitons, phonons and magnetism in correlated materials. This underscores the critical need for assumption-free, ab initio approaches to achieve an accurate description of magnetic excitons.

## Results

In our experimental studies, we investigate bulk CrSBr with magneto-optical spectroscopy, applying magnetic fields along the **c**-axis, which is the hard magnetisation axis. To gain comprehensive insight, we perform measurements at both low (0-3 T) and high (2-85 T) magnetic fields, providing a unique understanding of the excitonic response and exciton wavefunctions of CrSBr.

In the absence of an external magnetic field $B$, the Cr spins are aligned ferromagnetically within a layer, but antiferromagnetically between layers[19,24]. At -2 T and above, the magnetic field enforces the interlayer FM order, having an evident impact on the optical response. Figure 1a shows the representative spectra acquired with and without a $B$ = 2.5 T field, at temperature $T$ = 5 K. Two prominent excitonic features appear in the 1.3 − 1.9 eV spectral range in each case. We label them $X_A$ and $X_B$. At 0 T (blue curve), the transitions are located at 1.38 eV and 1.8 eV respectively. We note that the rich optical response around excitons $X_A$ and $X_B$ indicate that these are rather groups of closely-spaced transitions (as also supported by our theoretical calculations), probably further complicated by coupling of excitons with phonons[42] and photons[20,43]. However, it is the fundamental microscopic origin of the $X_A$ and $X_B$, their localized-delocalized nature, atomic and orbital character and symmetry, that determine how these

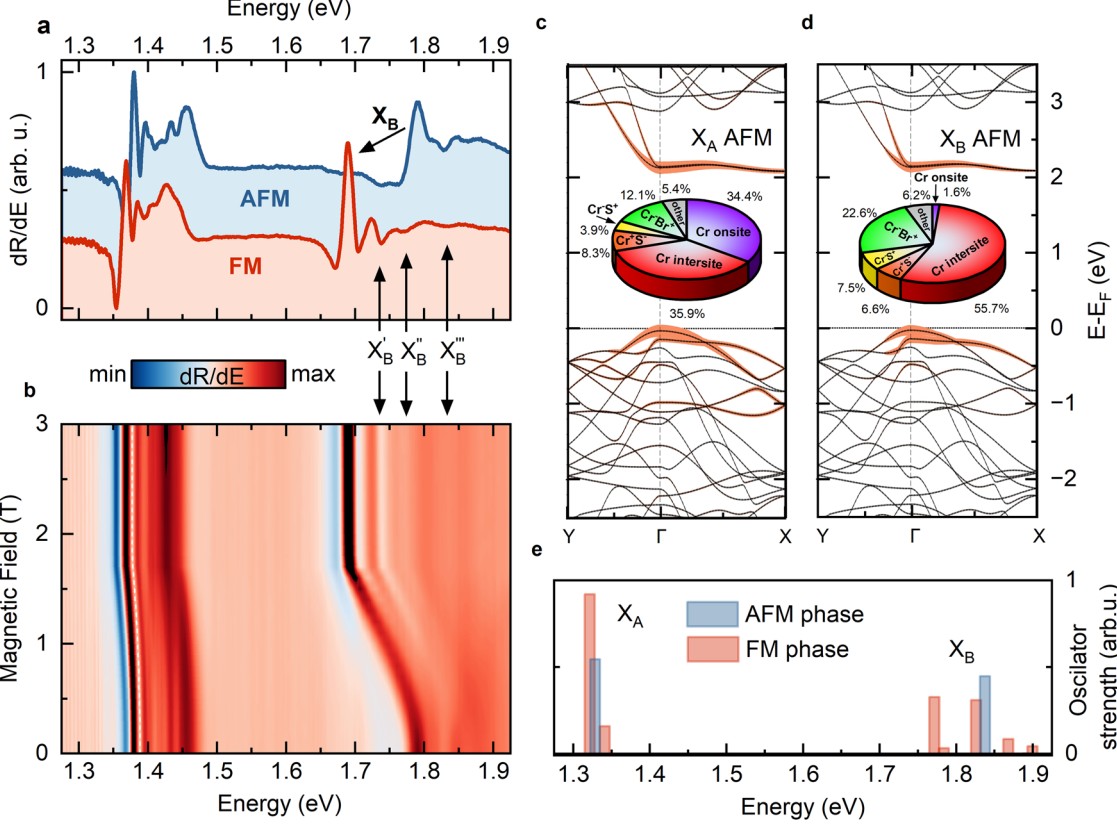

**Fig. 1 | Optical response and band structure of CrSBr. a** Low-temperature (-5 K) derivative of reflectivity spectra of bulk CrSBr in AFM phase (blue) and FM phase (red) induced by the magnetic field (2.5 T) applied along *c*-axis of the crystal (hard magnetisation axis)[59]. Arrows indicate the positions of two prominent excitonic features labelled $X_A$ and $X_B$. **b** Evolution of the reflectivity spectra derivative as a function of the magnetic field presented in false-colour maps. **c, d** Calculated CrSBr band structures in AFM phase with the decomposition of the $X_A$ and $X_B$ exciton wavefunctions in the band basis, highlighted by orange shading. The pie chart insets illustrate contributions of atomic orbitals to the exciton wavefunctions, showing a strong onsite Cr component for $X_A$. **e** Calculated oscillator strength of excitonic transitions in 1.3–1.9 eV energy range.

states couple with different bosons in the material. In the following, we use our diagrammatic parameter-free ab initio theory and magneto-optical studies at high and low fields to unambiguously explore the distinct microscopic origins of the $X_A$ and $X_B$ exciton states and how that difference reflects in their distinct coupling with magnetic field and lattice. An external magnetic field induces redshifts in both optical transitions, decreasing their energy with $B$ up to -1.8 T (Fig. 1b). Above 1.8 T, energy shifts of both excitons saturate, and exciton transition energies remain constant at around 1.37 eV and 1.7 eV respectively (Fig. 1a). This is a consequence of the magnetic ordering evolving continuously from AFM to FM between 0 and 1.8 T[19]. Note that the redshift is parabolic at low fields and symmetric in both positive and negative values of $B$, as expected.

Remarkably, the redshift of $X_B$, about 100 meV, is 10 times that of $X_A$. To explain this, we turn to the $QSG\widehat{W}$ framework we have developed[37,44,45]. This is a self-consistent, ab initio implementation of many-body perturbation theory (MBPT) that has been shown to give consistently high fidelity description of both the one-particle and two-particle properties of the electronic structure. As can be seen from Fig. 1(c) and (d), we find a band gap of approximately 2.07 eV, which is larger than the -1.5 eV reported in previous DFT-based $GW$ studies[19,46]. Another DFT-based $GW$ calculation was reported by Qian et al.[47]. This work, which used the plasmon pole approximation, estimated the gap to be 2.2 eV. Some reasons for the discrepancy in the different approaches are explained in the Methods section. Here, we note the recent photoemission experiments that place the band gap above 1.9 eV[48,49], in contrast to the initial prediction of 1.5 eV[50].

This larger value for the band gap has important consequences. A band gap of 2.07 eV implies that both $X_A$ and $X_B$ excitons are more strongly bound, and therefore more localised than was previously thought. Assuming the $QSG\widehat{W}$ gap of 2.07 eV is correct, $X_A$ and $X_B$ lie at 0.7 eV and 0.3 eV below the conduction band minimum, respectively. If the gap 1.5 eV, as previously calculated, both $X_A$ and $X_B$ would be Wannier-Mott-like[19,46]. Our results point to a more complex picture, namely, we find $X_A$ to have a significant Frenkel character and $X_B$ to be closer to the Wannier-Mott limit. 0.7 eV is too shallow for the $X_A$ exciton to approach the Frenkel limit, as is the case for CrX$_3$[23] and NiO[51], whose excitons are localised almost entirely to a single transition-metal-ligand molecular-orbital network (radius ~0.5 nm), nevertheless its Frenkel character is significant.

Further analysis provides detailed insight into the structure of both excitons. Pie chart insets in Fig. 1c, d reveal how the internal structure of the $X_A$ and $X_B$ differ: $X_A$ has a large onsite $dd$ component while $X_B$ does not. At the same time, $X_B$ has enhanced inter-site $dd$ and $pd$ dipolar character. $X_A$ spread over more energy states and a wider range of $\mathbf{k}$-space than $X_B$ see orange shading in Fig. 1c, d, representing the decomposition of the $X_A$ and $X_B$ wavefunctions in the band basis (see also extended discussion in SI about decomposition of excitonic states in band and orbital basis). The larger spread of $X_A$ over k-space directly reflects its localised nature in real space due to the Fourier relationship between momentum and position. Conversely, the localisation in $k$-space of $X_B$ is a hallmark of its enhanced spatial delocalisation. Note that the significantly real-space localised character also implies that a local atomic-orbital description, which is often used in ligand-field theory approaches, is a valid description of the $X_A$ and its associated transitions around 1.35 eV. We observe two clear peaks around 1.35, for both the transitions, the hole is primarily contained in the Cr-$d_{yz}$ orbital while the electrons come from the weakly split two conduction states of $d_{z^2}$ and $d_{x^2-y^2}$ character, respectively.

Importantly, the CrSBr excitons are neither of ideal Frenkel character nor of perfect Wannier-Mott character, contrasting with fully Frenkel excitons in CrBr$_3$ and fully Wannier-Mott excitons in MoS$_2$ as shown in Fig. S1 in SI. Having said that, the $X_A$ has more Frenkel character while $X_B$ has more Wannier-Mott character. This unique duality stems from the co-existence of a relatively small band gap (-2 eV) and

exciton binding energy, approximately 0.7 eV for $X_A$ and 0.3 eV for $X_B$, which places them in a fascinating intermediate regime between the classic Frenkel and Wannier-Mott limits. Note that in all 2D magnets it is natural to expect more Wannier-Mott-like excitons as we approach transitions close to the band edge with smaller binding energies (as was shown in previous works on CrX$_3$[23,52]). To put it in perspective, there are excitonic states in CrBr$_3$ at 1.3, 1.7, 2.0 eV of pure Frenkel character and at 3.0, 3.2 and 3.5 eV of Wannier-Mott character. This is natural since the band gap of CrBr$_3$ is 3.8 eV[53]. The significantly smaller band gap and relatively distinct exciton binding energies of $X_A$ and $X_B$ allows for the coexistence of excitons with mixed character in CrSBr; largely confined to a single Cr atom ($X_A$), and more delocalised excitons ($X_B$), which have significant intersite character (within a single layer). Further, CrSBr has two additional knobs for tuning its magneto-optical properties in a desired and controlled manner. CrSBr has large $ab$-planar anisotropy making both the Frenkel and Wannier transitions at 1.3 eV and 1.8 eV orient anisotropically along only $b$ direction (which can not happen in ideal hexagonal magnets) and its inter-layer AFM coupling that brings in an additional constraint on spin-hopping between layers, absent from magnets made out of FM layers only.

The contrasting structure of the $X_A$ and $X_B$ excitons explains the differences in their redshift in external magnetic field. Within the $QSG\widehat{W}$ framework, the energy of $X_B$ shift by 95 meV, and $X_A$ by 7 meV at $T=0$ K (Fig. 1e), in excellent agreement with the reflectance measurements. The microscopic mechanism behind this is as follows: in the AFM state, hybridisation between anti-aligned planes is spin-forbidden, creating an energy barrier for carriers at the band edges[19]. While in the FM state with spins aligned, the potential is uniform across the planes, reducing the energy barrier and consequently the band gap. Detailed calculations reveal an AFM band gap of -2.07 eV[48], and a 0.11 eV band gap reduction in FM phase. The $X_B$ exciton energy mostly tracks the conduction band, as expected for Wannier-Mott excitons. Thus, a reduction by 110 meV in the fundamental band gap results in a 95 meV reduction in the $X_B$ exciton energy. In contrast, the $X_A$ exciton, with its Frenkel-like character, exhibits a weaker dependence on the host band structure; the binding energy relative to a band edge state is less relevant in the ligand-field picture. Thus, the redshift in $X_A$ is much smaller, around 10 meV. In other words, the $X_B$ Wannier-Mott-like exciton is composed of states near the band edge, so its energy readily tracks any change in the host material's band gap. The highly localised and strongly bound $X_A$ exciton, however, is formed from a broad distribution of states across a large energy and momentum window. For this reason, $X_A$ is largely unaffected by the reduction in band gap that occurs in the FM phase. (see SI Fig. S2–S5 for the band and orbital decompositions of different excitonic states).

This remarkable difference observed for the $X_A$ and $X_B$ excitons, and the close correspondence with theory, has implications for the level of theory needed to understand excitons in CrSBr, and their interplay with the magnetic state. Excitons computed from the $G^{DFT}W^{DFT}$ approximation noted above[19,46] that predicts 1.5 eV gap, for example, would yield qualitatively inaccurate results. In the classical quantum chemical literature, Cr$^{3+}$ multiplet lines in Ruby are described by molecular ligand-field theory and the Tanabe-Sugano diagram[38,39] that originates from it. Sub-bandgap transitions in 2D magnets containing Cr$^{3+}$ ions are often phenomenologically interpreted in terms of transitions in the $D3$ Tanabe-Sugano diagram. At the opposite end of the spectrum lie the delocalised, non-magnetic, Wannier-Mott excitons, which are typically described by the Rydberg series[40,41] appropriate to Wannier excitons in $sp$ semiconductors. Surprisingly, in CrSBr, both kinds coexist, each with its own response to external perturbations, which cannot be adequately interpreted by these phenomenological models. In contrast, the ab initio $QSG\widehat{W}$ framework puts fermions and bosons on the same footing: its high fidelity has a predictive power that has been demonstrated in many kinds of systems. The single-particle spectrum $QSG\widehat{W}$, including the band gap and

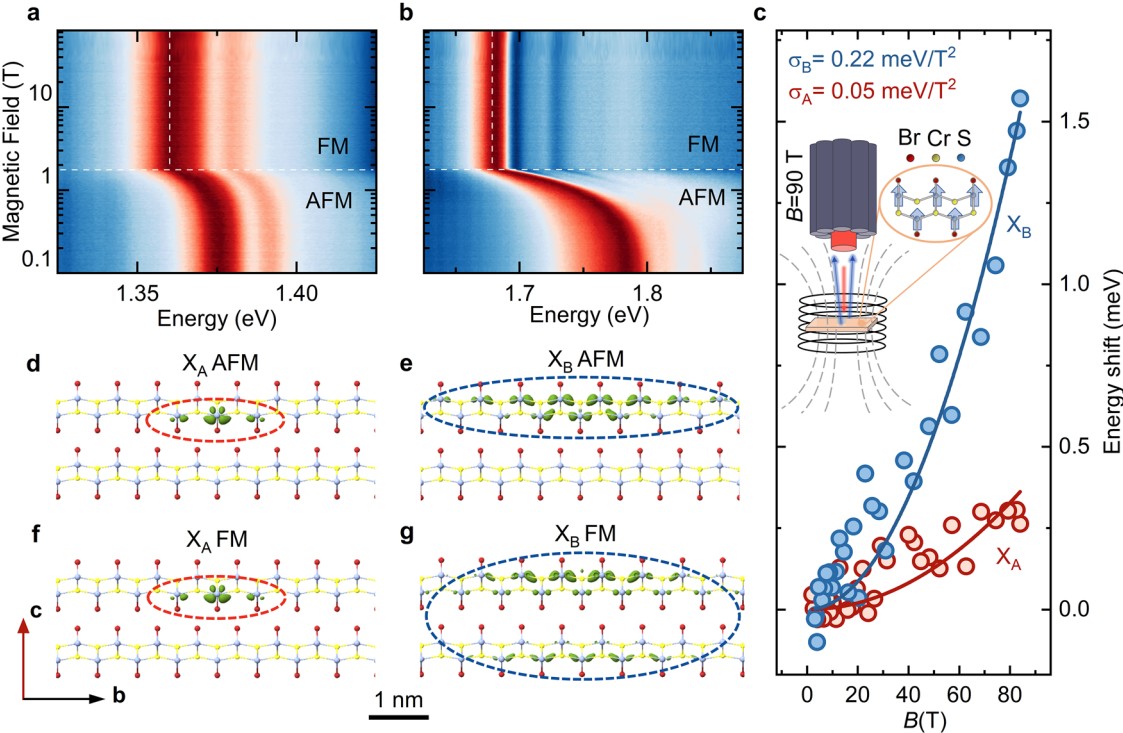

**Fig. 2 | High magnetic field studies of excitons wave function extensions.**
**a**, **b** Evolution of the reflectance spectrum as a function of magnetic field between 0 and 85 T. **c** Shifts in $X_A$ and $X_B$ transition energies measured at 2 K, as a function of the magnetic field in the FM phase, along with a parabolic fit, Eq. (1) with diamagnetic coefficients $\sigma_A = 0.05$ meV/T$^2$ and $\sigma_B = 0.22$ meV/T$^2$. The ratio of $\sigma_B/\sigma_A > 4$ confirms that $X_B$ is spatially more delocalised. The inset shows a schematic of the experimental setup where the CrSBr sample is placed in a coil of a pulse magnet. The optical fibre directs the broadband white light to the surface of the sample, and the reflected signal is collected by the surrounding bundle of fibre. **d**, **e** Isosurfaces for the $X_A$ and $X_B$ exciton wavefunctions overlaid on the crystal structure in the AFM phase, showing exciton confinement within a single layer. **f**, **g** Same as (**d**) and (**e**), but for the FM phase, revealing much stronger interlayer hybridisation for the $X_B$ exciton.

the role of disorder, shows good agreement with ARPES studies[48] as well as more recent studies along similar lines[49]. Furthermore, the excellent agreement with the optical response presented here and in other studies[26] provides another strong benchmark for the predictive power of this theory regarding two-particle electronic properties.

The distinct nature of $X_A$ and $X_B$ also accounts for the intriguing emergence of multiple excitonic features in the reflectance spectrum in the vicinity of $X_B$, indicated as $X'_B$, $X''_B$, and $X'''_B$ in Fig. 1a, b. As evident from panel (b), these new states gain oscillator strength with increasing $B$, enriching the optical response in the FM phase compared to the AFM phase. In contrast, the optical response associated with $X_A$ remains largely unaffected by the AFM-to-FM phase transition, aside from a gentle redshift of approximately 10 meV. This observation aligns with our $QSG\hat{W}$ calculations of the oscillator strength presented in Fig. 1e. At the AFM phase (blue bars), the optical response is dominated by two transitions. For $X_A$ This situation remains mostly unchanged in FM phase (see red bars). However, in the energy above $X_B$, several new transitions appear in the FM phase, as in the reflectance measurements.

The new transitions emerging at around 1.75 eV (see Fig. 1) stem from the multiplicity of valence and conduction states in CrSBr and from different parts of the Brillouin zone. Based on our analysis (see SI Fig. S2 and Fig. S4), these transitions exhibit some similarities to Wannier-Mott excitons in TMDs, where states centred around a high-symmetry point contribute to exciton formation by combining different valence and conduction bands[1,54]. However, an exciton with net zero momentum can be formed from electron and hole states belonging to the same $q$ point in the Brillouin zone even if the electrons and holes are not at any high symmetry point. Some of the transitions above 1.75 eV conform to that picture and, in that sense, they do not have an exact analogy in the Rydberg series. Having said that, it is true that in CrSBr, the presence of multiple valence and conduction states

within a narrow energy range (a few to several meV) facilitates a series of transitions within approximately ~50 meV around $X_B$ (See Fig. 1a, b). At the same time, the molecular character of $X_A$ and its strong localisation within a single layer in both AFM and FM phases result only in a minor impact with changes in spin confinement.

Finally, our decomposition of the exciton states into their band and orbital components (see Fig. 1, Fig. S2, and Fig. S4) demonstrates that the approximately 500 meV spectral separation between $X_A$ and $X_B$ does not arise from holes residing in different valence bands separated by a few hundred meV[46,55,56], but rather stems directly from their contrasting Frenkel and Wannier-Mott nature.

To gain further insight into the character and spatial extent of $X_A$ and $X_B$ excitons, we performed measurements under a magnetic field between 2 and 85 T at 2 K, where only FM order persists. The reflectivity spectra of both are presented in Fig. 2a, b, respectively. For both transitions, a blueshift in FM phase is observed, which value as a function of magnetic field is summarised in Fig. 2c (see also Fig. S9 in SI in narrower energy range). The data points follow a quadratic trend, which is ascribed to the diamagnetic shift of excitonic transitions[57]:

$$\Delta E = \sigma B^2 \qquad (1)$$

where the coefficient $\sigma$ is proportional to the expectation value of the squared radial coordinate perpendicular to **B** direction and reduced mass $\mu$:

$$\sigma = \frac{e^2}{8\mu}\langle r^2 \rangle \qquad (2)$$

Therefore, the observed blueshifts of both $X_A$ and $X_B$ can serve as a probe of the exciton wave function extension in the plane normal to

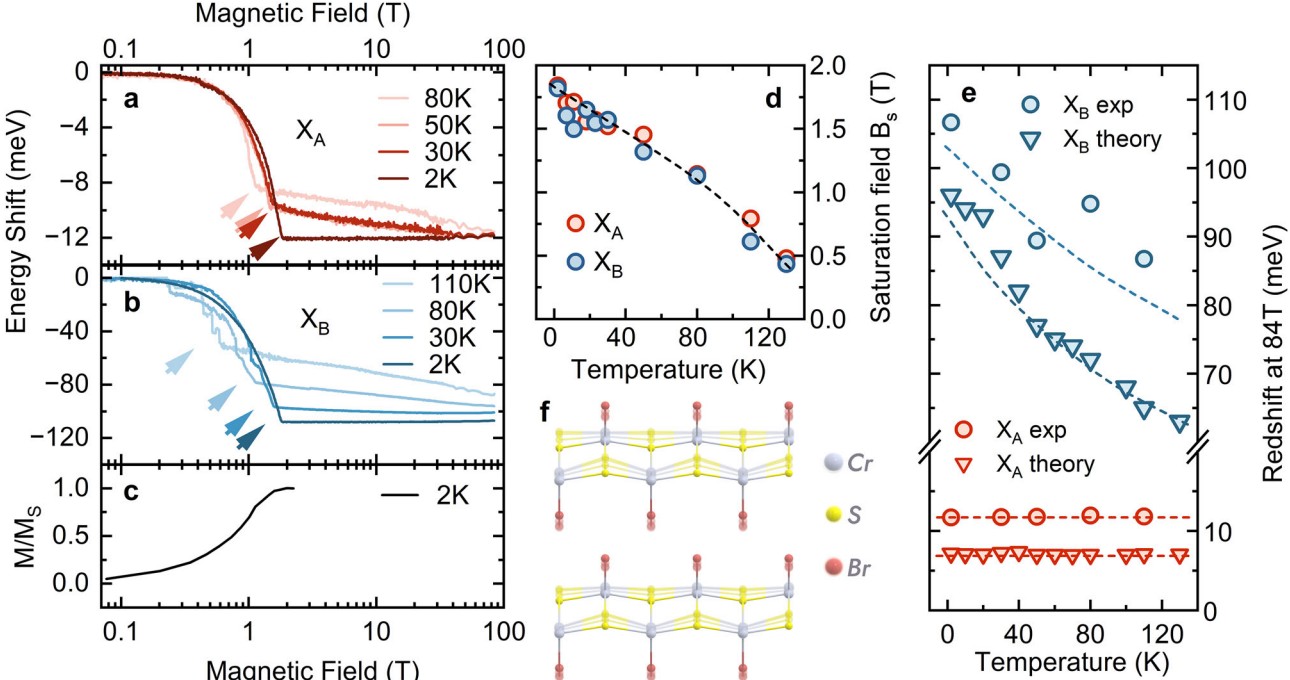

**Fig. 3 | Temperature dependence of magneto-optical response of CrSBr.**
**a**, **b** Shifts of the excitonic transition as a function of the magnetic field, measured at different temperatures for $X_A$ and $X_B$ (for linear scale see SI Fig. S8). Arrows indicate the inflexion (kink) points in the optical response related AFM-to-FM phase transition. **c** normalised magnetisation curve measured at 2 K, taken from ref. 58. **d** Dependence of the saturation field on temperature, extracted from optical spectra, with the dashed line serving as a guide to the eye. **e** Red and blue points represent absolute values of the energy shift between 0 T and ~85 T of $X_A$ and $X_B$ transitions as a function of temperature. Triangles are the results of the $X_A$ and $X_B$ energy shifts predicted by our $QSG\hat{W}$ calculations within the frozen-phonon approximation. The short and long-dash lines are guides to the eye (for theory and experiment, respectively). **f** The out-of-plane distortion of the lattice mediated by the $A_g$ phonon mode.

the magnetic field. We note that the diamagnetic shift coefficient should be understood as a measure of the effective exciton radius. While it doesn't resolve in-plane anisotropy, it remains a reliable tool for comparing the overall spatial extension of the two excitons. As shown in Fig. 2 the extracted shifts of $X_A$ and $X_B$ are fitted with Eq. (1). We find $\sigma_B$ (0.22 ± 0.02 meV/T$^2$) to be 4.4 times larger than $\sigma_A$ (0.05 ± 0.01 meV/T$^2$), an indication of its larger spatial extent.

This trend is reproduced in the theory. Figure 2d–g depict the isosurface of $X_A$ and $X_B$ excitonic wavefunctions for both AFM and FM phases (for the isosurface plots in other planes see SI Fig. S6 and S7). In the AFM phase, both excitons remain confined within a single layer and align along the $b$ direction. Since the inter-layer vdW coupling is AFM below 140 K, the hopping between layers is forbidden since electrons would need to tunnel or undergo a spin flip. However, either exciton can delocalise within a layer. One can notice that the different characters of two excitons have an impact on their interlayer extension in the FM phase. Since the in-plane and out-of-plane components of an exciton's wavefunction are intrinsically related to its binding energy, the strongly bound $X_A$ exciton retains its intralayer character as seen in Fig. 2f. In contrast, the more delocalized $X_B$ exciton extends to neighbouring layers in FM phase (Fig. 2g).

From the theory, we can compute the spatial extent of the individual excitons along the $b$ direction. We find $X_B$ and $X_A$ have lengths of 4.5 nm and 1.2 nm, respectively, with a ratio of 3.75 (corroborating that the spatial extension of $X_B$ is larger than $X_A$). Together, our high magnetic-field measurements and theory provide a new, unambiguous microscopic understanding of the $X_A$ and $X_B$ excitonic wavefunctions.

We turn to the temperature dependence of the optical response of the magnetic excitons. Figure 3a, b shows the energy shifts of the two excitonic states across a broad range of magnetic fields and temperatures. At low magnetic fields ( <2 T), the behaviour of both transitions qualitatively matches the direct magnetisation

measurements presented in panel (c) of Fig. 3. The initial redshift of $X_A$ and $X_B$, driven by spin canting relative to the $c$-axis, corresponds to a continuous magnetisation increase from zero to saturation. A characteristic kink in the energy shift (indicated by arrows) marks the saturation field ($B_S$) of magnetisation and the transition from the AFM to the FM phase.

Thermal fluctuations reduce both the saturation field $B_S$ and the magnetisation at the AFM-FM crossover[58,59]. Since the electronic structure is tied to the magnetic one, this effect directly manifests in the optical response. The magnetic field value corresponding to the kink, lowers with increasing temperature, as summarised in Fig. 3d. The temperature dependence of $B_S$ extracted from optical response measurements exhibits excellent agreement with direct magnetisation measurements and magnetoresistance investigations (see for instance[58,59]). This demonstrates that the fairly straightforward optical response of both exciton states can very effectively and directly probe the magnetic state or saturation field of CrSBr as a function of temperature.

The redshift of the excitonic transitions at the AFM-to-FM phase crossover (occurring at $B_S$) diminishes at higher temperatures owing to the reduced average spin alignment between neighbouring layers[20]. This thermal spin disorder can be suppressed by applying a sufficiently high magnetic field. As shown in panels (a) and (b) of Fig. 3, both excitonic transitions exhibit a gradual redshift in the FM phase at elevated temperatures, reflecting the progressive enhancement of spin alignment induced by the magnetic field. Notably, within the temperature range studied, the redshift of $X_A$ at high magnetic fields approaches the ~12 meV observed at 2 K (Fig. 3a). In contrast, the high field-induced redshift of the $X_B$ transition in the FM phase is temperature-dependent, decreasing at higher temperatures, though it still saturates in the high-field limit (see also SI Fig. S3). This distinct temperature dependence for $X_A$ and $X_B$ is summarised in (Fig. 3e). The $X_B$

redshift decreases from 110 meV at 2 K to ~85 meV at 110 K while the shift of $X_A$ remains almost temperature-independent, retaining its magnitude across the entire temperature range.

This difference further highlights the distinct nature of the two transitions and the differing impact of lattice vibrations on the two excitonic transitions. Given their different in-plane and out-of-plane wavefunction extensions, different exciton-phonon coupling is expected. For example, a significant Stokes shift, nearly absent for $X_A$, has been reported for $X_B$[60], suggesting strong exciton-phonon coupling in the latter, which is responsible for the temperature-dependent redshift of $X_B$ between the two magnetic orders, as shown below.

To verify this intuitive picture, we performed phonon calculations using phonopy[61,62] for CrSBr and computed the electronic and excitonic spectra using $QSG\widehat{W}$ under various lattice excursions along phonon eigenvectors of different symmetries and energies within the frozen-phonon approximation. The calculations assume full spin alignment in the FM phase. Similar analyses in various correlated magnetic and non-magnetic semiconductors[63–65] have provided invaluable insights into the fundamental interactions at play. Our theoretical phonon calculations reveal several $A_g$ modes (116, 237, 334 cm$^{-1}$) and $B_g$ modes (75, 76, 85, 174, 175, 178, 216, 292, 296, 339, 342, 359 cm$^{-1}$), consistent with previous theoretical[66] and experimental[67] studies.

The $B_g$ modes primarily cause in-plane lattice distortions, while the $A_g$ modes lead to out-of-plane distortions, for example, the displacement shown in Fig. 3f. Our computation shows a negligible impact of both types of vibrations on the $X_A$ redshift between the AFM and FM phases as shown by red triangles in Fig. 3e, in agreement with the experimental observations (see red circles). However, we find a remarkable impact of the $A_g$ modes (especially the $A_g^2$ mode at 237 cm$^{-1}$) on the renormalisation of the $X_B$ energy. These vibrations lead to distinct changes in $X_B$ energies in the two magnetic phases, and therefore, the AFM-to-FM redshift of this transition evolves strongly with temperature. As shown by the blue triangles in Fig. 3e, we obtained a very good, quantitative agreement between experiment and calculation. Note that the small discrepancy stems from the approximation that the temperature is extracted using a harmonic oscillator model, and also the fact that the temperature is fitted to a distortion created by a particular phonon eigenvector, while in reality, it's an ensemble involving all phonon modes. A more rigorous analysis of the temperature dependence can be performed using either an electron-phonon coupling theory or molecular dynamics calculations. However, the present approach, based on frozen phonon distortions along different phonon modes of distinct energies and symmetries, provides a key physical insight into the relevant lattice fluctuation mechanism. In contrast to the $A_g$ modes, the $B_g$ modes lead to similar corrections of $X_B$ energies in the AFM and FM phases. Consequently, they do not contribute to the observed temperature-dependent shift of the $X_B$ energy between these magnetic phases.

The results of this computation can be intuitively understood by considering the different characteristics of the wave functions of the two excitons. Despite $X_B$ being more extended in the $a$-$b$ plane than $X_A$ in both AFM and FM phases, the in-plane extent of $X_B$ does not change dramatically as the spins reorient, explaining the weak impact of in-plane modes. However, the out-of-plane extent of $X_B$ changes significantly in the FM phase (which is absent for $X_A$), thus $X_B$ has an inherent tendency to couple with out-of-plane lattice vibrations, which renormalise its energy.

## Discussion

Our study demonstrates that CrSBr bridges the gap between localised Frenkel and delocalised Wannier-Mott excitons, offering a unique platform to explore the interplay of excitonic effects, magnetism, and lattice dynamics in 2D semiconductors. By combining high-field magneto-optical spectroscopy with first-principles $QSG\widehat{W}$ calculations, we reveal the reasons behind the giant 100 meV redshift of $X_B$ excitons through the AFM-to-FM transition. The Wannier-Mott-like character and strong coupling to the band structure of this higher energy state make it a more sensitive probe of magnetic order than $X_A$ Frenkel-like exciton, by an order of magnitude. Furthermore, our temperature-dependent investigations highlight the important role of the exciton-phonon interactions, particularly out-of-plane vibrations, in the renormalisation of the Wannier-Mott-like states in CrSBr. The outstanding agreement between $QSG\widehat{W}$ and experimental findings (presented here and in other works[26,48]) allows us to conclude that both excitons are bound, with energies of 0.7 eV for $X_A$ and 0.3 eV for $X_B$. Crucially, our observations highlight that 2D magnetic materials like CrSBr can defy the conventional separation between Frenkel and Wannier-Mott excitons. The coexistence of two excitonic species with distinct real-space character, sensitivity to perturbations, and coupling to the lattice and spin degrees of freedom demonstrates that excitonic behaviour in this 2D magnetic material cannot be fully predicted by traditional phenomenological models. Therefore, assumption-free ab initio approaches with sufficient fidelity are required to reveal the excitonic landscape of these types of highly correlated materials.

## Methods

It has long been known that DFT-based $GW$, $G^{DFT}W^{DFT}$, provides a limited description of magnetic transition metal oxides, such as NiO[68], as well as many other correlated systems. Some adjustment to the starting point is essential; see for example ref. 69. However, the result depends on the choice of starting point, which causes ambiguity in the theory. The Quasiparticle Self-Consistent $GW$ approximation[44,45], QS$GW$, is a self-consistent form of Hedin's $GW$ approximation. Self-consistency removes the starting point dependence, and as a result, the discrepancies are much more systematic than conventional forms of $GW$.

However QS$GW$ has a tendency to overestimate band gaps slightly, particularly in oxides. The great majority of such discrepancies originate from the omission of electron-hole interactions in the RPA polarisability. By adding ladders to the polarizability, electron-hole effects are taken into account. Generating $W$ with ladder diagrams has important consequences; screening is enhanced and $W$ reduced. This in turn reduces fundamental band gaps and also valence bandwidths[37,70]. With the addition of ladder diagrams in $W$ ($QSGW \rightarrow QSG\widehat{W}$) this systematic overestimate is largely eliminated and $QSG\widehat{W}$ yields consistently high-fidelity band gaps and optical properties for a wide range of systems, including many magnetic insulators[37].

For a long time (and even today), the importance of self-consistency[70] was not widely appreciated, because historically, $GW$ has been mostly applied to weakly correlated $sp$ systems. There, $G^{DFT}W^{DFT}$ benefits from a fortuitous cancellation of errors: DFT has a tendency to underestimate band gaps, and the RPA has a tendency to underestimate $\epsilon_\infty$. These errors cancel to a great degree, leading to fortuitously good fundamental gaps. The fortuitous cancellation is far less effective in magnetic systems, in part because not only the DFT eigenvalues are poor, but the eigenfunctions are also.

This difficulty appears in the $G^{DFT}W^{DFT}$ studies noted earlier[19,46]. Its tendency to underestimate the band gap is consistent with many other studies of magnetic insulators. A subsequent $G^{DFT}W^{DFT}$ calculation[47] yielded a band gap in excess of 2.2 eV. The larger gap in this study is likely a result of the plasmon pole approximation used there, which has been shown to overestimate band gaps in oxides[71].

$QSG\widehat{W}$ differs in another essential way from DFT-based $GW$. The density is updated, including response to the magnetic field. $G^{DFT}W^{DFT}$ must rely on the DFT approximation not only to the density, but to the magnetisation and its dependence on the external field, which is also problematic.

For bulk CrSBr in the AFM phase with a 12-atom unit cell, we use $a = 3.504$ Å, $b = 4.738$ Å. Individual layers contain ferromagnetically polarised spins pointing either along the $+b$ or $-b$ axis, while the interlayer coupling is antiferromagnetic. Self-consistency for single particle hamiltonians (LDA, the static quasiparticlized QS$GW$ and $QSG\widehat{W}$ $\Sigma^0(\mathbf{k})$) are performed on a $10 \times 7 \times 2$ k-mesh while the relatively smooth dynamical self-energy $\Sigma(\mathbf{k}, \omega)$ is constructed using a $6 \times 4 \times 2$ k-mesh. The QS$GW$ and $QSG\widehat{W}$ cycles are iterated until the RMS change in $\Sigma^0$ reaches $10^{-5}$ Ry. A two-particle Hamiltonian for the BSE calculation of the polarizability, needed for both $\Sigma(\mathbf{k}, \omega)$ and the excitonic eigenvalues and eigenfunctions, contained 26 valence bands and 9 conduction bands. Excitonic eigenvalues of the two-particle Hamiltonian are converged using $10 \times 7 \times 2$ k-mesh.

## Sample synthesis

CrSBr crystals were made by the CVT method in a quartz ampoule directly from the elements. The ampoule ($40 \times 220$ mm) was filled with chromium (99.99%, -60 mesh, Chemsavers, USA), bromine (99.9999 %, Merck, Czech Republic) and sulphur (99.9999%, 2-6mm, Wuhan Tuocai Technology Co. Ltd., China) corresponding to 16 grams of CrSBr. Sulphur and bromine were used in 4 at.% excess. The ampoule was sealed under high vacuum using a diffusion oil pump and liquid nitrogen trap. The ampoule was placed in a crucible furnace and gradually, over a period of 4 days, heated on 700 °C, while the top of the ampoule was kept under 200 °C. Finally, the ampoule was placed in two two-zone horizontal furnace. First, the growth zone was heated to 900 °C and the source zone to 700 °C. Subsequently, the thermal gradient was reversed and the source zone was heated to 900 °C and the growth zone to 800 °C for ten days. Finally, the ampoule was cooled to room temperature and opened in argon-filled glovebox.

## Optical spectroscopy

The reflectance measurements in a high magnetic field were performed in a backscattering geometry. The bulk sample was placed in a liquid helium cryostat inside the coil of the pulsed magnet with the bore diameter of 4 mm. A tungsten halogen lamp providing a broadband white light source was guided to the sample by an optical fibre. The reflected light was collected by a fibre bundle, which surrounds the excitation fibre and is directed through into a 500 mm monochromator, equipped with a 300 gr/mm grating and a back-illuminated EMCCD camera. Spectra were captured in 1 ms intervals throughout the pulse duration (100ms). Measurements were conducted in two spectral ranges, around 900 nm ($X_A$) and 700 nm ($X_B$), repeatedly for each field range.

For the low magnetic field measurements, the bulk sample was mounted on the cold finger of a He flow optical cryostat equipped with 5 T superconducting magnet The reflectance measurements were performed in backscattering geometry with the use of a ×20 microscope objective (NA = 0.28). All these measurements were performed at 5 K. For the reflectance measurements, the white light was provided by a broadband halogen light source.

## Data availability

The magneto−optical data generated and/or analysed during the study are available without restrictions in the Zenodo database under the following online repository accesion code: https://doi.org/10.5281/zenodo.17941646.

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

## Acknowledgements

This work was authored in part by the National Laboratory of the Rockies for the U.S. Department of Energy (DOE) under Contract No. DE-AC36-08GO28308. For S.A., D.P., and MvS, funding was provided by the Computational Chemical Sciences programme within the Office of Basic Energy Sciences, U.S. Department of Energy. S.A., D.P., and M.v.S. acknowledge the use of the National Energy Research Scientific Computing Centre, under Contract No. DE-AC02-05CH11231 using NERSC award BES-ERCAP0021783 and we also acknowledge that a portion of the research was performed using computational resources sponsored by the Department of Energy's Office of Energy Efficiency and Renewable Energy and located at the National Laboratory of the Rockies and computational resources provided by the Oakridge leadership Computing Facility. The views expressed in the article do not necessarily represent the views of the DOE or the U.S. Government. The U.S. Government retains and the publisher, by accepting the article for publication, acknowledges that the U.S. Government retains a nonexclusive, paid-up, irrevocable, worldwide license to publish or reproduce the published form of this work, or allow others to do so, for U.S. Government purposes. Z.S and K.M. were supported by project LUAUS25268 from Ministry of Education Youth and Sports (MEYS) and by the project Advanced Functional Nanorobots (reg. No. CZ.02.1.01/0.0/0.0/15_003/0000444 financed by the EFRR). The publication was created as part of a project co-financed by the Polish Ministry of Science and Higher Education under contract no. 2025/WK/01.

## Author contributions

M.Ś. carried out all optical experiments, drafted the text and figures representing experimental results of the main manuscript and the supplementary information. M.R. participated in low magnetic field measurements and data processing, K.P. and P.Pe supported high magnetic field measurements. M.D. participated in data analysis and interpretation. D.P. contributed to the theoretical calculations. K.M. synthesised the CrSBr crystal with the support and supervision of Z.S. M.v.S. contributed to the theoretical calculation and manuscript writing. F.D. was involved in data interpretation and manuscript writing. M.B., together with P.P. supervised magnetic field measurements, participated in data analysis interpretation, and manuscript writing. S.A. performed theoretical calculations, helped in interpreting the observations, conceived the main theme of the work and contributed to manuscript writing.

## Competing interests

The authors declare no competing interests.

## Additional information

**Peer review information** : *Nature Communications* thanks Yu Ye and the other anonymous reviewer(s) for their contribution to the peer review of this work. A peer review file is available.

