## [Transparent Peer Review file · Nature Communications]

Distinct Magneto-Optical Response of Frenkel and Wannier Excitons in CrSBr

Corresponding Author: Dr Paulina Plochocka

Version 0:

Reviewer comments:

Reviewer #1

(Remarks to the Author)

This manuscript systematically investigates the excitonic properties in the 2D magnetic semiconductor CrSBr through a combination of theoretical and experimental approaches, revealing the coexistence of localized Frenkel-like excitons (XA) and delocalized Wannier-Mott-like excitons (XB), as well as their distinct responses to magnetic perturbations and lattice vibrations. The work is innovative and scientifically sound, providing important insights into the exciton-magnetism coupling mechanisms in 2D magnetic materials. Overall, the data are solid and the conclusions are credible, and I recommend accepting this manuscript after addressing the following issues.

(1) The current title, "Unraveling the nature of excitons in the 2D magnetic semiconductor CrSBr," is somewhat broad. Given that the core finding of the paper is the duality of the two excitons and their distinct magnetic responses, it is suggested that the title be made more specific, for example, by highlighting key features such as "Frenkel-like and Wannier-Mott-like excitons" or "their distinct magneto-optical responses."

(2) There have been recent studies on XB excitons in CrSBr, with some conclusions differing from those in this paper (e.g., the view that XB excitons originate from transitions from split valence bands to the conduction band minimum). It is recommended that the authors supplement the discussion of these literatures, clarify the differences between the conclusions of this paper and other studies, and explain the reasons, so as to strengthen the contextual relevance of the research.

(3) The authors mention that "the binding energy relative to a band edge is less relevant in the ligand-field picture" but do not clearly explain why XA only shifts by ~10 meV in magnetic fields (while the band redshift reaches 110 meV). It is necessary to further elaborate on the microscopic mechanism behind the weak response of XA to magnetic fields, combining its Frenkel-like characteristics.

(4) The authors state that "An exciton with zero momentum can be formed from electron and hole states belonging to the same q point in the Brillouin zone even if the electrons and holes are not at any high symmetry point." It is necessary to further verify the scientific validity of this statement theoretically, for example, by supplementing explanations based on the momentum conservation condition of excitons.

(5) The authors explain that XA is localized due to the forbidden interlayer transitions in the antiferromagnetic phase, but do not clarify why XA remains interlayer-localized in the ferromagnetic phase. It is necessary to supplement the localization mechanism of XA in the ferromagnetic phase (such as the influence of wavefunction characteristics or interlayer coupling).

(6) Fig. 3b shows non-monotonic jumps in XB energy during the AFM-to-FM phase transition at high temperatures. It is recommended that the authors analyze the physical origin of this phenomenon (e.g., the influence of temperature-induced lattice vibrations or spin fluctuations).

(7) In Fig. 3e (temperature-dependent redshift) and Fig. 2c (magnetic field-induced blueshift), there is a superposition effect of diamagnetic blueshift and redshift caused by suppressed thermal perturbations. It is recommended that the authors clarify the quantitative relationship between these two effects and explain the accuracy of the experimental data and the sources of errors.

(8) For example, the statement "the shift of XA remains temperature-independent" is not rigorous and should be revised to "the energy shift of XA is almost temperature-independent after spin fluctuations are suppressed by a strong magnetic field." Such expressions need to be corrected to ensure scientific accuracy.

(9) There are typos in the text (e.g., "For XA This situation"). It is recommended that the authors proofread the entire manuscript to ensure accurate expression.

In summary, it is recommended that the authors supplement and revise the above issues. The paper can be published after further improvement.

Reviewer #2

(Remarks to the Author)

In the manuscript "Unraveling the nature of excitons in the 2D magnetic semiconductor CrSBr" by Smiertka and et al., the authors demonstrate that CrSBr hosts both localized Frenkel-like and delocalized Wannier-Mott-like excitons. These two types of excitons in CrSBr exhibit strikingly different responses to magnetic field and lattice perturbations. More specifically, the Wannier-Mott-like XB exciton is more sensitive to magnetic order changes, has a larger spatial extent, and exhibit a stronger temperature dependent redshift.

The discovery reported in this manuscript is interesting. However, I think the connections between different sections are not tight enough. Especially, both the redshift and the exciton-phonon coupling are related to out-of-plane spread, while the section "Exciton wavefunctions in high magnetic field" only probes the in-plane spread. Additionally, several very relevant recent works are not being reviewed in this manuscript (e.g. exciton-phonon coupling <https://doi.org/10.1021/acsnano.3c07236>, GW calculation [10.1103/PhysRevResearch.5.033143](https://doi.org/10.1103/PhysRevResearch.5.033143)).

Comments and questions:

1. The authors mentioned that "CrSBr bridges the two distinct pictures of Wannier-Mott and quasi-Frenkel excitons, providing a new platform to investigate the interplay between different excitonic regimes ...". Could the authors explain why CrSBr can have this uncommon feature? In addition, the hybridization feature of the XA exciton in monolayer CrSBr has also been reported ([10.1103/PhysRevResearch.5.033143](https://doi.org/10.1103/PhysRevResearch.5.033143)).
2. There are multiple excitonic peaks close to XA and XB in Fig 1(a, b). Why does this work only focus on XA and XB? What about other excitonic peaks?
3. The authors mentioned that in ref. [26] and ref. [51], the DFT-based GW method gives a smaller bandgap. However, there are also GW calculations with above 2eV bandgap values ([10.1103/PhysRevResearch.5.033143](https://doi.org/10.1103/PhysRevResearch.5.033143)). Please compare all these methods.
4. When analyzing the internal structure of the XA and XB excitons, the authors only show the excitons in the AFM phase. As the excitonic states change with magnetic order, the exciton structures in the FM phase should also be showed and compared.
5. The manuscript mentioned that "XA is spread over more energy states and a wider range of k-space than XB". What is the physical meaning of this feature? How does it related to the localized or delocalized feature of XA and XB excitons? For Cr intersite, is it at the same layer or different layers? BTW, it should be "XA spread over ..." instead of "XA is spread over ..."
6. The manuscript mentioned that "In contrast, the XA exciton, with its Frenkel-like character, exhibits a weaker dependence on the host band structure". As both exciton have wavefunctions on the conduction bands shown in In Fig 1(c) and 1(d), how to see the weak dependence from the wavefunction decomposition?
7. Eq. (1) and Eq. (2) are used to estimate the in-plane spread of XA and XB excitons in the FM states. However, as the magnetic field is beyond 2T, the above experiments and estimations cannot measure the spread of XA and XB excitons in the AFM states. Additionally, as the dielectric response of CrSBr is very anisotropic, I think it is necessary to show the isosurface of XA and XB in both a and b directions.
8. The authors use Fig.2 to show that the in-plane radius of XB is a few times larger than XA. However, I think this is not adequate to explain large redshift in XB exciton as the AFM-FM transition induced redshift is related to the out-of-plane spread.
9. I assume the isosurface of XB and XA would be very anisotropic. Then how to calculate the radius? Is it for a fixed isosurface? If so, it should be mentioned.
10. The blue shift is impossible to see in Fig. 2(a) and 2(b).

Reviewer #3

(Remarks to the Author)

The authors report reflectance measurements on CrSBr without and with magnetic fields up to 85 Tesla and compare their results with simulated data from DFT-GW plus ladder diagrams which allow for the computation of excitons. They identify two excitons, one rather tightly bound and one only weakly bound. Their differing responses to parameter changes such as magnetic field and/or temperature are detected and discussed.

This is very nice, solid work on excitons and I appreciate this study of the

interplay of different degrees of freedom which surely is very important in CrSBr. Also the collaboration of experiment and theory is an asset. The obtained agreement is good and helps to understand the data.

Yet, I do not see any particular reason to publish this work in a highlight journal. The fact that two different bound states react differently is rather expected. As far as I see the theory captures only the FM and the AFM phase, not the canting in the applied field. Phonons are only considered in a frozen manner. The results for the sigma coefficients show a ratio of 4.4, which does not match to the ratio of the lengths which is 3.75 because the lengths enter quadratically in the sigma factors. The progress in the theoretical simulations has been published before in the articles by van Schilfgaarde et al.

In conclusion, I recommend publication in a more specialized journal as regular article.

Version 1:

Reviewer comments:

Reviewer #1

(Remarks to the Author)

All my raised concerns are now cleared in the revision. I support the publication of the manuscript in Nature Communications.

Reviewer #2

(Remarks to the Author)

In the reply, the authors give clearer explanations from theoretical or computational aspect, but make no further attempt to clarify questions related to their experiments. As this is a combined theoretical and experimental study, I think the experimental part is not tightly connected with the theoretical part. Moreover, the presence of two types of excitons has previously been reported in other calculations. With these two reasons in mind, I think this work is suitable for a more specialized journal.

Reviewer #3

(Remarks to the Author)

I thoroughly read the extensive rebuttal to all the questions raised by the Referees. In essence, the rebuttal boils down to emphasizing that a tightly bound exciton is local in real space and thus extended in reciprocal space. Then, it does not depend on band edges but rather on the changes of the band as a whole, for instance on the average energy over the entire band. Conversely, a weakly bound exciton is extended in real space, but fairly localized in reciprocal space. Then, it is strongly susceptible to the position of the band edges.

Frenkel and Wannier excitons are well known and established. So the novelty seems to lie merely in the fact that they occur in the same material. From the theoretical point of view, this is not so special. Any box potential can easily be tuned to a range where one bound state is tightly bound while a second is very close to the continuum of scattering states.

Of course, for a real material reliable ab initio computations are a remarkable achievement. But the response in the rebuttal clearly underlines that the method has been introduced before and has been performed for over 100 magnetic systems.

The response to the second Reviewer's question where the multiple peaks come from remains very vague: of course, several bands and/or phonon assisted transitions may play a role. But please be definite: What does the theory tell us? Are the many X_A 's and X_B or only one each?

The statements in the rebuttal on zero momentum of the exciton are strongly misleading. In order to achieve zero momentum from two constituents, they must have opposite momenta (or in case of particles and holes the same momenta) - the velocities are not important.

In summary, I stick to my recommendation that this is nice and publishable work.
But I do not see the level of novelty warranting publication as Nature Communication.

Version 2:

Reviewer comments:

Reviewer #1

(Remarks to the Author)

I would like to express my sincere gratitude for the constructive discussion that transpired between the reviewers and the authors, centering on the novelty and significance of this scholarly endeavor. As a researcher who has worked on CrSBr, I appreciated the submitted work. Consequently, I reaffirm my original position and support the publication of this work.

Reviewer #2

(Remarks to the Author)

As the authors provide no further experimental results to distinguish the in-plane feature vs the out-of-plane feature (The related experiments are mentioned in my first reply), I stay with my previous opinion.

Version 3:

Reviewer comments:

Reviewer #1

(Remarks to the Author)

I found that Reviewer #2 had a different opinion regarding the evaluation of the manuscript. Based on my own judgement, however, I stand by my original position and support the publication of this work.

At the beginning, we would like to sincerely thank the referees for their time and effort in reviewing our manuscript and for providing a comprehensive list of constructive questions and suggestions, which helped us to improve our work. Below we present a detailed response to all referees' questions.

Reviewer #1 (Remarks to the Author):

This manuscript systematically investigates the excitonic properties in the 2D magnetic semiconductor CrSBr through a combination of theoretical and experimental approaches, revealing the coexistence of localized Frenkel-like excitons (XA) and delocalized Wannier-Mott-like excitons (XB), as well as their distinct responses to magnetic perturbations and lattice vibrations. The work is innovative and scientifically sound, providing important insights into the exciton-magnetism coupling mechanisms in 2D magnetic materials. Overall, the data are solid and the conclusions are credible, and I recommend accepting this manuscript after addressing the following issues.

(1) The current title, "Unraveling the nature of excitons in the 2D magnetic semiconductor CrSBr," is somewhat broad. Given that the core finding of the paper is the duality of the two excitons and their distinct magnetic responses, it is suggested that the title be made more specific, for example, by highlighting key features such as "Frenkel-like and Wannier-Mott-like excitons" or "their distinct magneto-optical responses."

We have changed the title to the following: "*Distinct Magneto-Optical response of Frenkel-like and Wannier-Mott-like excitons in CrSBr*"

(2) There have been recent studies on XB excitons in CrSBr, with some conclusions differing from those in this paper (e.g., the view that XB excitons originate from transitions from split valence bands to the conduction band minimum). It is recommended that the authors supplement the discussion of these literatures, clarify the differences between the conclusions of this paper and other studies, and explain the reasons, so as to strengthen the contextual relevance of the research.

This is a good question, and an answer requires several separate discussions. One part has to do with prior interpretations in terms of ligand-field theory. This is a key point of our paper, and we enlarged the discussion explaining why ligand-field theory is more appropriate for the X_A exciton than the X_B exciton. Another concerns the fidelity of the hamiltonian used to generate the excitonic structure. Reviewer II raised this point as well. The methods section had some discussion about why the QSGW theory we use surmounts deficiencies in other approaches, for example, why an underestimated bandgap leads to wrong conclusions, especially with regard to the X_B exciton.

We have augmented that discussion a little (see our answer to Review II):

"This difficulty appears in the GDFTW DFT studies noted earlier [19, 46]. Its tendency to underestimate the bandgap is consistent with many other studies of magnetic insulators. A subsequent GDFTW DFT calculation [47] yielded a bandgap in excess of 2.2 eV. The larger gap in this study is likely a result of the plasmon pole approximation used there, which has been shown to overestimate band gaps in oxides [70]. QSGW differs in another essential

way from DFT-based GW. The density is updated, including the response to the magnetic field. GDFTW DFT must rely on the DFT approximation not only to the density, but to the magnetisation and its dependence on the external field, which is also problematic.”

Finally, this question points to the difficulty in visualising the excitons in the band basis. For excitons with Frenkel-like character, this is not trivial. We explain below why this is so in some detail, in the following paragraphs. A truncated version is presented in the paper.

In a system where electrons and holes attract each other weakly, the exciton delocalizes over large length scales in real space. In contrast, if the Coulomb attraction between the electrons and holes is extremely strong, both the electron and the hole that form the exciton wave function can get localised on the same atom. Typical Wannier excitons fall in the first category, and Frenkel excitons fall in the second category.

The question is now how to visualise them in the band basis. Chemists have historically used the ‘atomic multiplets’ language to describe the atom-local excitations (aka Frenkel excitations). Because such transitions are atom-local, Chemists would often describe these states with energy dynamics on the atom involving different atomic orbitals. This picture, then, would not need any momentum description of the states. For several decades, this has been a remarkably successful interpretation of Frenkel exciton states in Ruby and other large band-gap magnetic systems. These two-particle transitions are atom-local, and they don’t see the band-information of the host system.

The same language also works for several deep lying exciton states in CrX_3 series since they have large band gaps and excitons with binding energies of 2-3 eV and can be described as orbital-orbital transitions on the same atom. Similar descriptions face limitations in systems like CrSBr (and several others), where the band gap is not too large ~ 2 eV and exciton binding energies are smaller (in Ruby binding energy is ~ 5 eV, in CrCl_3 binding energy is 3 eV). The excitons in CrBrS have some Frenkel character and also some extended, Wannier-Mott character at the same time. They do respond to changes in the band gap and momentum-dependent changes in the host band structure. Our work shows both experiment and theory, needed to probe, understand, and control such excitonic states.

Most importantly, what we are able to show is that for the X_A state, where the Frenkel language should hold to a large extent, an orbital description of the excitations is, indeed, feasible. Cr is in a t_{2g}^3 configuration. It is expected that the low energy optical transitions should involve occupied t_{2g} and unoccupied e_g states. We observe that X_A contains two sub-peaks. For both transitions, the hole is primarily contained in the Cr- d_{yz} orbital, while the electron comes from the bands with d_z^2 orbital character for X'_A and $d_{x^2-y^2}$ character for X''_A . At the conduction edge at Γ point, these two bands with distinct orbital characters are weakly split and lead to these two transitions around 1.3 eV. In the original draft, when we used the term ‘split bands,’ this is what we primarily meant. Further, note that d_{yz} orbital character on the valence side is shared by more than one low-energy band, and also the orbital character is spread over several momentum points in the Brillouin zone. All such k points that satisfy the matching-group-velocity criterion to form an exciton, take part in the exciton formation. Note that we showed such band-basis visualisation of Frenkel excitons in the first for the excitons in CrX_3 in our previous publications. A figure showing this is also included in the SI. We also show how the CrX_3 excitons (Frenkel), MoS_2 excitons (Wannier) appear very

different in the band basis and how the CrBrS excitons are intermediate to those two extreme limits.

In strong contrast to X_A , the transitions around the X_B mostly emerge from the valence and conduction edges close to Γ point and have strong Wannier-Mott character.

Importantly, our analysis shows that X_B involves states related to d_{yz} orbitals and some of S-p character (two top-most valence bands around Γ point), however, the contribution of the next valence band, separated by around ~ 200 meV is absent. Therefore, the difference in the spectral position of X_A and X_B does not stem from the contribution of deeper valence bands to X_B but from their contrasting characters (more Frenkel or more Wannier-like), and it is in that sense, our analysis and understanding of these excitons are different from some other publications in the literature.

This is expected since our approach is closest to the Hedin's prescribed exact solution of the many-body problem, in strong contrast to commonly used LDA-based single-shot GW approaches. Contrary to that, ours is a self-consistent approach which does not depend on the starting point (LDA, DFT potentials or Hartree-Fock) and also treats the interplay between the excitonic correlations and electronic self-energy self-consistently, a mechanism that is entirely missing from traditional LDA-GW approaches. The remarkable success of our theory (in completely getting rid of any free parameters, particularly, U , which is often used in LDA-GW calculations for magnets) in predicting excitonic features of about 100 magnets have been shown in several reviews and regular articles. Our theory predicts the band gap of the system to be 2 eV in strong contrast to the LDA-GW theories that predicted it at 1.5 eV. It was this severe underestimation of band gap from LDA-GW theories that led to the wrong conclusion of Wannier character of X_A exciton, and also leads to several other wrong conclusions, including the band decomposition of the exciton states. Note that our predicted ~ 2 eV band gap (and, consequently, large binding energy for X_A and small binding energy for X_B) was recently verified in an ARPES study by Watson et al. that was able to carefully remove the charging effects to probe the intrinsic states of CrSBr. A recent paper on CrSBr uses a theory closer in spirit to ours and finds the band gap at 1.95 eV (Nature Communications volume 16, Article number: 1134 (2025)), naturally agreeing with our estimations.

We have discussed the complexity of visualising the excitons in the band basis in the revised version of the supporting information. We also added several paragraphs

discussing this at length in the revised manuscript and SI (following the referee's suggestion).

“(see also extended discussion in SI about decomposition of excitonic states in band and orbital basis). The larger spread of X_A over k -space directly reflects its localised nature in real space due to the Fourier relationship between momentum and position.”

“This unique duality stems from the co-existence of a relatively small bandgap (~ 2 eV) and exciton binding energy, approximately 0.7 eV for X_A and 0.3 eV for X_B , which places them in a fascinating intermediate regime between the classic Frenkel and Wannier-Mott limits...”

“In other words, the X_B Wannier-Mott-like exciton is composed of states near the band edge, so its energy readily tracks any change in the host material's...”

“Finally, our decomposition of the exciton states into their band and orbital components (see Fig. 1, S2, and S4) demonstrates that the approximately 500 meV spectral separation between X_A and X_B does not arise from holes residing in different valence bands...”

We thank the referee for this question, as it allows us to address a very pertinent and nontrivial issue regarding these excitons and which enhances the quality of our paper.

(3) The authors mention that "the binding energy relative to a band edge is less relevant in the ligand-field picture" but do not clearly explain why X_A only shifts by ~ 10 meV in magnetic fields (while the band redshift reaches 110 meV). It is necessary to further elaborate on the microscopic mechanism behind the weak response of X_A to magnetic fields, combining its Frenkel-like characteristics.

The answer to this question follows the reasoning presented in our response to the previous comments. Excitons form from electrons (conduction) and holes (valence) from certain momenta and energies. Excitons in non-magnetic semiconductors, for example, in black Phosphorus, GaAs and several TMDs, the electron and hole states forming the excitons come from the band edges and from a narrow momentum window. An exciton that forms from the band edges typically tracks the motion of the band edge, and this is observed for several Wannier-Mott excitons, where, as the band gap changes (due to the number of layers, strain), the excitons show the same changes. This is exactly what happens for the X_B exciton. X_B is weakly bound (binding energy of ~ 0.3 eV) compared to the more Frenkel-like X_A (strongly bound, localised with binding energy 0.7 eV). X_B excitons, as we have shown, form primarily from the band edges, while X_A forms from multiple states distributed over a large energy window and momentum points across the Brillouin zone. Hence, X_A remains responsive less strongly to the changes in the band gap, which is a typical feature of Frenkel-like localised excitons. When the AFM-FM transition is induced by an external magnetic field, the band gap changes by ~ 100 meV, and X_B is able to track the change of the band gap (as it is predominantly built from the band edge state), while X_A can not.

We clarified this in the revised manuscript:

“...the binding energy relative to a band edge state is less relevant in the ligand-field picture. Thus, the redshift in X_A is much smaller, around 10 meV. In other words, the X_B Wannier-Mott-like exciton is composed of states near the band edge, so its energy readily tracks any change in the host material’s bandgap. The highly localised and strongly bound X_A exciton, however, is formed from a broad distribution of states across a large energy and momentum window. For this reason, X_A is largely unaffected by the reduction in bandgap that occurs in the FM phase.”

(4) The authors state that "An exciton with zero momentum can be formed from electron and hole states belonging to the same q point in the Brillouin zone even if the electrons and holes are not at any high symmetry point." It is necessary to further verify the scientific validity of this statement theoretically, for example, by supplementing explanations based on the momentum conservation condition of excitons.

We thank the reviewer for bringing this point to our attention and gladly elaborate further. Excitons with zero momentum can be formed if the hole and electron group velocities are the same. This can be understood as a condition in which the electron and the hole are able to move together as a bound pair. The requirement of equal group velocities implies that the exciton can be formed whenever the gradient of conduction and valence band dispersions is equal, which is the case for high symmetry points, but can also be satisfied in other points of the Brillouin zone (a good example is C exciton in TMDs).

In a situation when excitons are in the intermediate limit and span over a large k-space, this condition is even softened as the combination of different states can result in zero-net momentum.

(5) The authors explain that XA is localized due to the forbidden interlayer transitions in the antiferromagnetic phase, but do not clarify why XA remains interlayer-localized in the ferromagnetic phase. It is necessary to supplement the localization mechanism of XA in the ferromagnetic phase (such as the influence of wavefunction characteristics or interlayer coupling).

The X_A exciton is a strongly bound Frenkel-like exciton with a small radius. A key explanation behind this is covered mainly in answers to questions (2) and (3). In brief, because it is tightly bound, it remains strongly localised even when the inter-layer spin hopping restriction is relaxed in the FM phase.

We attach a figure here for the X_A exciton in the FM phase with an isosurface upper bound cut at only 1%. As can be seen, there is a weak (almost negligible) leakage of the wavefunction into the next layer. This reveals the key ingredient of the XA exciton: the large binding energy of this exciton in both the AFM and FM phases is able to (largely) localise the exciton inside a single layer even when the inter-layer spin hopping constraint is relaxed.

We added according comment to the manuscript when discussing wave function extension (Fig 2(d)-(g)):

“One can notice that the different characters of two excitons have an impact on their interlayer extension in FM phase. Since the in-plane and out-of-plane components of an exciton’s wavefunction are intrinsically related to its binding energy, the strongly bound X_A exciton retains its intralayer character as seen in Fig. 2 (e). In contrast, the more delocalized X_B exciton extends to neighbouring layers in FM phase (Fig. 2(g)).”

(6) Fig. 3b shows non-monotonic jumps in X_B energy during the AFM-to-FM phase transition at high temperatures. It is recommended that the authors analyze the physical origin of this phenomenon (e.g., the influence of temperature-induced lattice vibrations or spin fluctuations).

We thank the reviewer for this observation. The apparent non-monotonic behaviour of the X_B exciton energy during the AFM-to-FM phase transition is due to experimental accuracy, especially in the low field limit, where the magnetic field and so the exciton energy change rapidly. We emphasise that this data was acquired in a pulse magnetic field where the whole magnetic pulse duration is ~ 300 ms, which provides a significant experimental challenge. The data acquired in a static magnetic field (up to 3T) presented in the initial part of the manuscript are free of such artefacts. Additionally, this effect is visually exaggerated by the logarithmic x-axis scale in Fig. 3b.

The same dataset is plotted on a linear scale in the Supplementary Information (Fig. S5), where the energy shift appears smooth and the small deviations are no longer pronounced.

(7) In Fig. 3e (temperature-dependent redshift) and Fig. 2c (magnetic field-induced blueshift), there is a superposition effect of diamagnetic blueshift and redshift caused by suppressed thermal perturbations. It is recommended that the authors clarify the quantitative relationship between these two effects and explain the accuracy of the experimental data and the sources of errors.

We appreciate the reviewer’s suggestion and agree that both the diamagnetic blueshift and the redshift due to the interplay between magnetic order and thermal disorder may contribute to the total exciton energy evolution. However, these effects occur on very different energy

scales (tens of meV vs single meV), with the redshift dominating over the much smaller diamagnetic contribution at high temperatures.

Our focus in the manuscript is on the overall trends, rather than the precise decomposition of overlapping contributions. A detailed, quantitative disentangling of these effects would require a more targeted study, which we consider beyond the scope of the present work.

To support our interpretation, below we show a plot of the X_B exciton energy vs magnetic field at 80 K, both with and without the estimated diamagnetic shift subtracted (based on the 2 K curve). The negligible difference between the two confirms that the dominant contribution is the redshift, and the overall trend is not affected.

(8) For example, the statement "the shift of XA remains temperature-independent" is not rigorous and should be revised to "the energy shift of XA is almost temperature-independent after spin fluctuations are suppressed by a strong magnetic field." Such expressions need to be corrected to ensure scientific accuracy.

We agree with the referee and have corrected the statement accordingly.

(9) There are typos in the text (e.g., "For XA This situation"). It is recommended that the authors proofread the entire manuscript to ensure accurate expression. In summary, it is recommended that the authors supplement and revise the above issues. The paper can be published after further improvement.

We have revised the manuscript to remove this issue.

Reviewer #2 (Remarks to the Author):

In the manuscript “Unraveling the nature of excitons in the 2D magnetic semiconductor CrSBr” by Smiertka and et al., the authors demonstrate that CrSBr hosts both localized Frenkel-like and delocalized Wannier-Mott-like excitons. These two types of excitons in CrSBr exhibit strikingly different responses to magnetic field and lattice perturbations. More specifically, the Wannier-Mott-like X_B exciton is more sensitive to magnetic order changes, has a larger spatial extend, and exhibit a stronger temperature dependent redshift.

The discovery reported in this manuscript is interesting. However, I think the connections between different sections are not tight enough. Especially, both the redshift and the exciton-phonon coupling are related to out-of-plane spread, while the section “Exciton wavefunctions in high magnetic field” only probes the in-plane spread. Additionally, several very relevant recent works are not being reviewed in this manuscript (e.g. exciton-phono coupling <https://doi.org/10.1021/acsnano.3c07236>, GW calculation 10.1103/PhysRevResearch.5.033143).

The essential results of our present work can be interpreted in terms of the differences in the binding energies of the X_A and X_B . X_A has a large binding energy and strong Frenkel character and X_B has a small binding energy and Wannier-Mott character. The difference shows up in the ab-plane spatial extension of the two excitons. Our fully ab-initio theory, combined with the field-dependent studies, explores this fundamental difference between the X_A and X_B (and relates it to the observed redshift between AFM and FM phases). This difference between X_A and X_B is independent of the AFM or FM nature of the magnetic state and, hence, is the core argument presented in our work.

The fact that spin hopping constraints between the layers relax going from the AFM to the FM phase, and that leads to the remarkable redshift of the X_B (and not X_A) is also down to that fundamental difference in excitonic character, Frenkel and Wannier. The subsequent strong coupling of X_B with out-of-plane phonon modes (and not so much with the X_A) is also a testimony to the distinct localised-delocalised character of the excitons. With the help of our theory, we are able to capture the intriguing nature of the inter-layer delocalisation and coupling of X_B with out-of-plane phonon modes.

We completely agree with the referee; in an ideal situation, we would like our experimental magneto-optics capabilities to map out the out-of-plane extension of the X_B states, too. However, this is a challenging task and experimental estimation of the out-of-plane spread would require future studies with alternative experimental approaches, like changing the crystal axis probed by the light beam, which would be experimentally very challenging and is beyond the scope of this work.

However, our experimental temperature-dependent studies, combined with theory, provide a systematic understanding of all essential observations, while the experimental mapping out of X_B 's out-of-plane delocalisation remains an open issue, and we believe our work will motivate further experiments in this field.

We also thank the reviewer for pointing out the missing references. We have updated the manuscript accordingly, including the suggested references and comments on the

relation between in-plane size, out-of-plane size and binding energy in the revised manuscript:

“One can notice that the different characters of two excitons have an impact on their interlayer extension in FM phase. Since the in-plane and out-of-plane components of an exciton’s wavefunction are intrinsically related to its binding energy, the strongly bound X_A exciton retains its intralayer character as seen in Fig. 2 (e). In contrast, the more delocalized X_B exciton extends to neighbouring layers in FM phase (Fig. 2(g)).”

Comments and questions:

1. The authors mentioned that “CrSBr bridges the two distinct pictures of Wannier-Mott and quasi-Frenkel excitons, providing a new platform to investigate the interplay between different excitonic regimes ...” Could the authors explain why CrSBr can have this uncommon feature? In addition, the hybridization feature of the XA excitation in monolayer CrSBr has also been reported (10.1103/PhysRevResearch.5.033143).

We thank the reviewer for these extremely important questions. There are two parts to the answer:

1. For any magnetic insulator, there are some excitons which have large binding energies and Frenkel character, while there would always be some weakly bound excitons with Wannier character. For example, in CrBr_3 , there are 1.3, 1.7, 2.0 eV excitons with significant Frenkel character, and then there are 3.2 eV and 3.5 eV excitons with significant Wannier character. This is natural since the band gap of CrBr_3 is ~ 3.8 eV. But this large band gap and large exciton binding energies largely separate out Frenkel states from Wannier states very clearly. A small band gap of 2 eV and exciton binding energies of 0.7 and 0.3 eV delicately place CrBrS in an intermediate regime.
2. Further, in terms of tunability of excitonic states, what makes CrBrS special is its anisotropy and the inter-layer AFM coupling. The anisotropy is so strong that both 1.3 and 1.8 eV transitions are clearly polarised along the b axis, and there are no concomitant transitions along the a axis, making their detection and tuning along one axis easier. Further, the inter-layer coupling being AFM allows us an additional knob that is the magnetic field that can drive an AFM-FM transition, allowing for spin-hopping between layers, which is otherwise restricted in the AFM phase. Note that CrX_3 , being FM and hexagonal, misses out on both degrees of freedom.

We provided more extensive discussion about Frenkel and Wannier-Mott characters of excitons in SI, as well as we point out the difference in the revised version of the manuscript:

“This unique duality stems from the co-existence of a relatively small bandgap (~ 2 eV) and exciton binding energy, approximately 0.7 eV for X_A and 0.3 eV for X_B , which places them in a fascinating intermediate regime between the classic Frenkel and Wannier-Mott limits...”

2. There are multiple excitonic peaks close to XA and XB in Fig 1(a, b). Why does this work only focus on XA and XB? What about other excitonic peaks?

We agree with the referee's observation that the derivative of the reflection spectrum in Fig. 1(a) and (b) indeed points to a rich excitonic landscape around 1.3 eV and 1.7-1.8 eV.

This observation, in fact, strongly supports our theoretical predictions. As shown in our theoretical analysis (Fig. 1e and SI Fig. S4), we are not dealing with a single transition around X_A and X_B , but rather a group of closely-spaced transitions involving different valence and conduction bands. Furthermore, we expect the optical response to be enriched by phonon-assisted transitions (phonon replicas) of the excitonic features, as previously reported [ACS Nano 2024, 18, 4, 2898–2905].

While a precise, line-by-line identification of every transition is beyond the scope of this work, we emphasise that the primary conclusion of our study—the distinct Frenkel-like and Wannier-Mott-like character of the two exciton families—remains robust. The existence of multiple transitions within each group does not weaken the fundamental distinction we have established between the two types of excitons and their drastically different magneto-optical responses.

We provided the following comment to the manuscript:

“We note that the rich optical response around excitons X_A and X_B indicate that these are rather groups of closely-spaced transitions (as supported further by our band structure calculation), probably further complicated by phonon-replica [43].”

3. The authors mentioned that in ref. [26] and ref. [51], the DFT-based GW method gives a smaller bandgap. However, there are also GW calculations with above 2eV bandgap values (10.1103/PhysRevResearch.5.033143). Please compare all these methods.

We thank the referee for bringing this paper to our attention. We now include a reference to this Phys Rev Research paper and concur with several of its statements. We also comment in the Methods section, that the use of the plasmon pole approximation in this paper is the likely reason why this particular DFT-GW calculation yields a larger gap than its antecedents, purporting to use the same method. We also add comments to the manuscript in general why GW based on DFT has several limitations. Our rebuttal to Reviewer 3 also explains the contrast between our approach and DFT-based GW approaches.

The following comments were added to the manuscript:

“Another DFT-based GW calculation was reported by Qian et al. [47]. This work, which used the plasmon pole approximation, estimated the gap to be 2.2 eV. Some reasons for the discrepancy in the different approaches are explained in the Methods section. Here, ...”

“A subsequent $G^{DFT}W^{DFT}$ calculation [47] yielded a bandgap in excess of 2.2 eV. The larger gap in this study is likely...”

4. When analyzing the internal structure of the X_A and X_B excitons, the authors only show the excitons in the AFM phase. As the excitonic states change with magnetic order, the exciton structures in the FM phase should also be showed and compared.

We have now included in the Supplementary Information all the figures from both the AFM and FM phases showing the real-space structure of X_A and X_B along all three directions and also their atomic decompositions.

5. The manuscript mentioned that “ X_A is spread over more energy states and a wider range of k-space than X_B ”. What is the physical meaning of this feature? How does it related to the localized or delocalized feather of X_A and X_B excitons? For Cr intersite, is it at the same layer or different layers? BTW, it should be “ X_A spread over ...” instead of “ X_A is spread over ...”

We thank the reviewer for this insightful question. Regarding the physical interpretation, the fact that X_A spreads over a broader range of energy states and k-points indicates that it is composed of contributions from a wide portion of the Brillouin zone. This directly reflects its localised nature in real space, due to the Fourier relationship between momentum and position space. A wavefunction that is localised in momentum space is spread over real space. This is exactly the scenario with the Wannier-Mott excitons that we are used to seeing in non-magnetic semiconductors and all TMDs (see Fig.S1). In this case, the excitons are usually spread over several nanometers in real space, which also implies that they only form from a very narrow momentum window, mostly from the band edges.

Conversely, an exciton that is highly localised in real space should delocalise over several momentum points over the Brillouin zone. Further, these localised transitions can be thought of as onsite dd transitions involving two d orbitals. When visualised in the band basis, an orbital character can spread over several bands. Naturally, a localised Frenkel exciton emerges from several momentum states and bands spread out over a large energy window (electron and hole states of certain orbital characters, wherever they are in the momentum and energy). We showed this and established the paradigm rigorously for the Frenkel excitons observed in CrX_3 in our previous works. We also revise the SI information where we provide detailed descriptions of two exciton states in band and orbital basis. We provide further details and insights derived from this analysis in reply to referee’s question 6.

The term Cr intersite refers to Cr atoms in the same layer. We added additional comments to the manuscript to clarify these aspects:

“The larger spread of X_A over k-space directly reflects its localised nature in real space due to the Fourier relationship between momentum and position space. Conversely, the localisation in k-space of X_B is a hallmark of its enhanced spatial delocalisation.”

“This allows for the coexistence of excitons largely confined to a single Cr atom (X_A), and more delocalized excitons (X_B), which have significant intersite character (within a single layer).”

6. The manuscript mentioned that “In contrast, the X_A exciton, with its Frenkel-like character, exhibits a weaker dependence on the host band structure”. As both exciton have wavefunctions on the conduction bands shown in In Fig 1(c) and 1(d), how to see the weak dependence from the wavefunction decomposition?

Excitons form from electrons (conduction) and holes (valence) from certain momenta and energies. Excitons in non-magnetic semiconductors, particularly interesting are the examples of excitons in black Phosphorus, GaAs and several TMDs, the electron and hole states forming the excitons come from the band edges and from a narrow momentum window. An exciton that forms from the band edges typically tracks the motion of the band edge, and this is observed for several Wannier-Mott excitons, whereas the band gap changes (due to the number of layers, strain), the excitons track the changes. This is exactly what happens for the X_B . X_B is weakly bound (binding energy of only 0.3 eV) compared to the more Frenkel-like X_A (strongly bound, localised with binding energy 0.7 eV). X_B , as we have shown below, forms primarily from the band edges, while X_A forms from multiple states distributed over a large energy window and momentum points distributed over the Brillouin zone. Hence, X_A remains less affected by the changes in the band gap, which is a typical feature of Frenkel-like localised excitons. When the AFM-FM transition happens with the magnetic field, the band gap changes by ~ 100 meV, and X_B is able to track the change in the band gap (as it is predominantly built from the band edge state), while X_A can not. For example, we observe that X_A has two substructures. For both transitions, the hole is primarily contained in the Cr- d_{yz} orbital, while it is the weakly split two conduction states of d_z^2 and $d_{x^2-y^2}$ character. This orbital character causes X_A to be spread over many states in a band basis. In strong contrast, the substructures around the X_B transition mostly emerge from the valence and conduction edges close to Γ point and have strong Wannier-Mott character.

We provided an extended discussion about these aspects in SI and also clarified this in the revised manuscript:...

“...the binding energy relative to a band edge state is less relevant in the ligand-field picture. Thus, the redshift in X_A is much smaller, around 10 meV. In other words, the X_B Wannier-Mott-like exciton is composed of states near the band edge, so its energy readily tracks any change in the host material's bandgap. The highly localised and strongly bound X_A exciton, however, is formed from a broad distribution of states across a large energy and momentum window. For this reason, X_A is largely unaffected by the reduction in bandgap that occurs in the FM phase.”

7. Eq. (1) and Eq. (2) are used to estimate the in-plane spread of X_A and X_B excitons in the FM states. However, as the magnetic field is beyond 2T, the above experiments and

estimations cannot measure the spread of X_A and X_B excitons in the AFM states. Additionally, as the dielectric response of CrSBr is very anisotropic, I think it is necessary to show the isosurface of X_A and X_B in both a and b directions.

We thank the reviewer for this important point. In response to the second part of the comment, we now show isosurfaces of the real-space exciton wavefunctions for both X_A and X_B, resolved along the a and b crystallographic directions, in the revised Supplementary Information, to better illustrate their anisotropic spatial profiles.

We agree that probing the diamagnetic shifts of the excitons in the antiferromagnetic (AFM) phase would provide valuable insight. However, due to the very low critical field required for the AFM-to-ferromagnetic (FM) transition, our current experimental approach is unable to reliably measure diamagnetic shifts in the AFM phase. The required high magnetic fields inevitably induce this phase transition, limiting our measurements to the FM state.

Probing the exciton spatial extension in the AFM phase would indeed necessitate a different experimental approach, and we consider this an interesting and important direction for future investigation.

Despite this limitation, our high-field measurements in the FM phase provide clear evidence for the distinct character of the two excitons, X_A and X_B. This experimental finding is strongly supported by our theoretical calculations, which further demonstrate that the different Frenkel and Wannier-Mott characters of these excitons are preserved in both the AFM and FM phases. On a side note, often perturbing a material in a certain state is necessary to gain information about its original state. However, sometimes even weak perturbations can change the state, but in the adiabatic limit, that perturbed state still contains the information of the original state. The weak magnetic field is enough to drive an AFM-FM transition, but still provides invaluable information from the original AFM state that remains robust across the perturbed FM phase as well.

8. The authors use Fig.2 to show that the in-plane radius of X_B is a few times larger than X_A. However, I think this is not adequate to explain large redshift in X_B exciton as the AFM-FM transition induced redshift is related to the out-of-plane spread.

We have shown the exciton wavefunctions and their extensions along different crystalline directions in the figures in the revised SI. There are two observations: that the X_B remarkably redshifts by 100 meV with a magnetic field when the AFM-FM transition happens, and that X_B is able to delocalize between layers in the process. However, the shift of 100 meV can be simply explained based on the reduction of the band gap by ~100 meV when the AFM-FM transition happens and the fact that the X_B is primarily a band edge exciton, which is able to track the motion of the band edge going from the AFM to FM. Which brings us back to the key argument of our paper, which is that X_A is fundamentally different from X_B, irrespective of AFM or FM magnetic states. The relaxation of spin-hopping between the layers in FM phase leads to additional structures to the X_A and X_B wavefunctions compared to their AFM phase. What we carefully attempted to do in this work is to pin down the cause-and-effect for various observations. While for interpreting some observations, the ability of X_B to extend between layers is crucial for most other observations, the fundamental difference (Frenkel and Wannier) between X_A and X_B (irrespective of their AFM or FM phases) is the key mechanism.

We will clarify this distinction in the revised manuscript to better connect the in-plane measurements to the out-of-plane exciton behaviour.

“One can notice that the different characters of two excitons have an impact on their interlayer extension in FM phase. Since the in-plane and out-of-plane components of an exciton’s wavefunction are intrinsically related to its binding energy, the strongly bound X_A exciton retains its intralayer character as seen in Fig. 2 (e). In contrast, the more delocalized X_B exciton extends to neighbouring layers in FM phase (Fig. 2(g)).”

9. I assume the isosurface of X_B and X_A would be very anisotropic. Then how to calculate the radius? Is it for a fixed isosurface? If so, it should be mentioned.

We thank the reviewer for this important remark. Indeed, the isosurfaces of X_A and X_B excitons are anisotropic, reflecting the elliptical shape of their wavefunctions due to the anisotropic electronic and dielectric structure of CrSBr, as shown by the isosurface in the manuscript and revised version of SI.

Note that, theoretically, mapping out the wavefunction along all directions is not a difficult task, and we are able to do that. However, experimentally, it is a challenge. The radii extracted from our diamagnetic shift measurements represent the expectation value of the squared radial coordinate perpendicular to the B direction (as explained in the manuscript). Therefore, the extracted value should be understood as an average or effective radius, which does not resolve the in-plane anisotropy, but still allows for comparing the spatial extension of two excitons. This is where the theory is able to help – theory shows that the wavefunction extension along a direction barely changes, meaning the primary difference emerges from the difference in extension along the b axis.

To better illustrate the actual shape of the exciton wavefunctions, we included anisotropic isosurfaces in the SI. We will also clarify in the manuscript that the diamagnetic shift coefficient is an effective measure of wave function expansion and does not directly reflect the anisotropic extent in a and b directions.

“We note that the diamagnetic shift coefficient should be understood as a measure of the effective exciton radius. While it doesn’t resolve in-plane anisotropy, it remains a reliable tool for comparing the overall spatial extension of the two excitons”

10. The blue shift is impossible to see in Fig. 2(a) and 2(b).

Indeed, the diamagnetic blueshift is on the order of 1 meV, and therefore not visible on the 100 meV energy scale used in Fig. 2(a) and 2(b). To address this, we have now included zoomed-in versions of the relevant spectral regions in the SI, where the small but measurable blueshifts can be seen by eye.

Reviewer #3 (Remarks to the Author):

The authors report reflectance measurements on CrSBr without and with magnetic fields up to 85 Tesla and compare their results with simulated data from DFT-GW plus ladder diagrams which allow for the computation of excitons.

They identify two excitons, one rather tightly bound and one only weakly bound. Their differing responses to parameter changes such as magnetic field and/or temperature are detected and discussed.

This is very nice, solid work on excitons and I appreciate this study of the interplay of different degrees of freedom which surely is very important in CrSBr. Also the collaboration of experiment and theory is an asset. The obtained agreement is good and helps to understand the data.

Yet, I do not see any particular reason to publish this work in a highlight journal. The fact that two different bound states react differently is rather expected.

We thank the reviewer for the positive feedback on our work. We would like to clarify why our specific findings, while seemingly straightforward, are a crucial and non-trivial contribution that makes the work highly suitable for a high-impact journal.

The reviewer correctly notes that different bound states are generally expected to react differently to external perturbations. However, the unprecedented and highly significant finding of our work is the **coexistence of two fundamentally distinct exciton species—quasi-Frenkel and Wannier-Mott-like—within the same material**. This is a rare phenomenon in which these two exciton types form from a single, common electronic band structure.

Our work goes beyond simply observing a difference in behaviour. We provide the first microscopic understanding of this unique phenomenon, demonstrating an intriguing interplay between the exciton's nature and the host band structure that is of paramount importance for the resulting magneto-optical response. Furthermore, our findings defy conventional models. The simultaneous existence of these two exciton types highlights the limitations of traditional phenomenological theories—such as molecular ligand-field theory (for Frenkel excitons) and the Rydberg series (for Wannier-Mott excitons)—in fully describing the complex interplay of excitons, phonons, and magnetism. Our study thus underscores the critical need for the advanced *ab initio* many-body perturbation theory presented in the paper. Further, it also addresses key issues pertaining to two different scientific communities trying to understand and probe magnetic excitonic states – the chemistry community uses the classical Frenkel picture with atom-local transitions (atomic multiplets), on the opposite extreme is the solid-state-physics community that uses Rydberg series to analyse extended excitons in traditional semiconductors. What we reveal, first, is that the excitons in these 2D magnets are neither of the two extremes –rather, are mostly intermediate. In a step change, we show that it is this relative strength of the Frenkel and Wannier characters that changes as the exciton binding energy changes in 2D magnets, and that tells us how tunable these states are and what perturbations should be used to tune them. Our key achievement is that we are able to combine high-field measurements and advanced many-body theory (completely free of parameters and model assumptions) to map out the internal structure of these excitonic states unambiguously.

In summary, the key contribution of our work is not just observing a difference, but delivering a microscopic understanding of that difference, proving that a single material can host two such disparate excitonic states with vastly different magneto-optical responses. We are convinced that the still-growing community working on this particular material and other 2D magnets will benefit significantly from our insights and experimental confirmations on the nature of excitons in CrSBr.

As far as I see the theory captures only the FM and the AFM phase, not the canting in the applied field. Phonons are only considered in a frozen manner.

We could definitely do a canted-spin analysis showing intermediate steps and also density functional perturbation theory for electron-phonon coupling. However, doing calculations for the intermediate steps would not reveal additional information (beyond the broad scope as we have already established in our work by probing the limiting cases) for the distinct microscopic nature for these exciton states and their tunability. Further, we could do an electron-phonon coupling theory analysis to include the full phonon ensemble to explore their impact on exciton energies. However, in the frozen-phonon approach, we scan through the entire eigenvector space of phonons individually to explore how each of them impacts the exciton energies. This allows us to explore the key mechanism and the ability to extract information for the important phonon modes that are coupled with excitonic states based on symmetry preferences. Such clean insight would be rather hard (not impossible, though) to extract from a fully electron-phonon coupling theory.

The results for the sigma coefficients show a ratio of 4.4, which does not match to the ratio of the lengths which is 3.75 because the lengths enter quadratically in the sigma factors.

The discrepancy in diamagnetic shift scaling is expected, and values extracted from iso-surfaces and diamagnetic shift should be considered only in a qualitative manner. We do not claim that the experimentally estimated 3.75 factor should be compared against the 4.4 factor computed from the theory. The radii extracted from our diamagnetic shift measurements represent the expectation value of the squared radial coordinate perpendicular to the field direction (as explained in the manuscript). Therefore, the extracted value should be understood as an average or effective radius, which does not resolve the in-plane anisotropy, but still allows for comparing the spatial extension of two excitons. Most importantly, we note that the expression traditionally used to extract the radii from diamagnetic shift measurements is only rigorously true for isotropic excitons and not for highly anisotropic excitons as observed in CrBrS. The generalisation of that expression is still absent from the literature and is up for an outstanding undertaking for future scientific work.

The progress in the theoretical simulations has been published before in the articles by van Schilfgaarde et al.

On a side remark, our theory is not LDA-GW but is completely different. Our theoretical approach is a self-consistent QSG \hat{W} framework, which solves the Hedin's equations self-consistently within a quasi-particle approximation. There are two key features that makes our theory stand out compared to regularly used LDA-GW approaches:

1. The self-consistency removes the starting point dependence; our electronic and excitonic spectra are independent of what starting point is used, LDA, any DFT potential, Hartree-Fock or hybrid-functional.
2. We solve the Bethe-Sapleter equations to extract the optical polarizability and the excitons, but this is not a one-step process, as is the case with any commonly available LDA-GW codes. In the presence of the excitonic correlations, we recompute the self-energy and charge densities (back and forth) until all the quantities, the excitonic correlations, W , G , Σ and charge density reach self-consistency. This is extremely crucial in magnetic systems where the excitonic correlations are large and they can significantly impact the band structural properties, a mechanism entirely missing from LDA-GW single-shot approaches.

The impact of these key ingredients of our theory are remarkable in magnetic systems. We have published review articles (Computer Physics Communications 249, 107065) and several works (Physical Review B 108 (16), 165104) showing examples of around 100 magnetic systems where our theory remarkably improves the predictive capabilities for electronic and excitonic spectra in magnets. This is also important for CrBrS. Most of the initial publications on CrBrS predicted a band gap at 1.5 eV and concluded that the excitons were weakly bound Wannier excitons based on LDA-GW. This, as we can establish now, is a wrong conclusion and would be in conflict with the remarkable differences observed for the X_A and X_B excitons. We have published six papers on CrSBr over the past two years using our theory, and this recent work, as we believe, is the most substantial one that combines theory and experiment to unambiguously explore the magneto-optical tunability of these exciton states.

We hope that the referee will find our publication suitable for the journal.

Response to Second Referees Report

Reviewer #1

All my raised concerns are now cleared in the revision. I support the publication of the manuscript in Nature Communications.

We are grateful to the Reviewer for the positive opinion about revised version of the manuscript.

Refereeing to the previous suggestion of referee we further modify title of our manuscript to following one: "*Distinct Magneto-Optical response of Frenkel and Wannier excitons in CrSBr*"

Reviewer #2

In the reply, the authors give clearer explanations from theoretical or computational aspect, but make no further attempt to clarify questions related to their experiments. As this is a combined theoretical and experimental study, I think the experimental part is not tightly connected with the theoretical part. Moreover, the presence of two types of excitons has previously been reported in other calculations. With these two reasons in mind, I think this work is suitable for a more specialized journal.

We have a strong objection to the reasoning presented by Referee #2, who claims that "*the experimental part is not tightly connected with the theoretical part*". We strongly contend that this assertion fundamentally mischaracterises our work and the nature of the research. The described measurements and their results served to empirically probe the spatial extension of different excitonic states' wave functions. **Crucially, these are the identical wave functions that were calculated with the presented theory.** Hence, the connection between experiment and theory is straightforward. The **measured observables directly validate the theoretical predictions** regarding exciton character and spatial extent. We argue that this represents an indispensable connection where the two elements are mutually dependent for the overall scientific conclusion. We stress that the extension of the exciton wave function is a crucial and definitive indicator of the exciton character (Wannier-Mott vs. Frenkel). Therefore, **our experiments provide empirical support for the theory, and the theory, in turn, explains the observed experimental results about strength of different exciton states coupling to magnetic order.**

The second argument to reject our manuscript, presented by Referee #2, is "*the presence of two types of excitons has previously been reported in other calculations.*" While we acknowledge that the general concept of localised and delocalized excitons exists in theoretical literature, we must emphasise that our work constitutes a critical and novel advancement for the following reasons:

Our study provides the first comprehensive and combined experimental and theoretical characterisation of simultaneously existing states Frenkel and Wannier-like excitons in CrSBr. As detailed above, the true scientific novelty lies in the unprecedented synergy where theory and experiment are tightly linked to reveal the exciton character. We transition from a theoretical concept to empirical proof and exhaustive physical description in this highly fascinating material. Moreover, our paper delivers a crucial, detailed understanding of the magneto-exciton coupling in CrSBr, identifying key facts and dependencies that the scientific community was previously unaware of. **This is not a mere re-reporting of a concept; it is the first realization and validation of this physics in CrSBr.**

Taking all the above into consideration, the argumentation used by Referee #2 to reject the paper is scientifically unwarranted.

Reviewer #3

I thoroughly read the extensive rebuttal to all the questions raised by the Referees. In essence, the rebuttal boils down to emphasizing that a tightly bound exciton is local in real space and thus extended in reciprocal space. Then, it does not depend on band edges but rather on the changes of the band as a whole, for instance on the average energy over the entire band. Conversely, a weakly bound exciton is extended in real space, but fairly localized in reciprocal space. Then, it is strongly susceptible to the position of the band edges.

We are pleased the Referee agrees with the one of the core conclusion of our work: the distinction between the real-space localization/reciprocal-space extension of excitons and its direct consequence on their dependence on the band structure. This fundamental observation was indeed a strong emphasis in our previous rebuttal and the revised manuscript. However, **we wish to emphasize that the novelty of our work extends significantly beyond this conclusion. It serves to explain the unique magneto-optical properties of CrSBr. Specifically, we highlight that our work is the only one to provide the microscopic origin for the contrasting magneto-exciton coupling observed for two exciton species.** We not only identify the existence of different exciton types within a single material but also connect their distinct nature to magneto-exciton coupling and exciton-phonon coupling. This comprehensive analysis offers a crucial understanding of the unique properties of CrSBr, understanding that the scientific community currently lacks.

We stress that all of this context, along with these conclusions regarding the coupling mechanisms, has been already thoroughly integrated and detailed within the previously revised version of the manuscript.

Frenkel and Wannier excitons are well known and established. So the novelty seems to lie merely in the fact that they occur in the same material. From the theoretical point of view, this is not so special. Any box potential can easily be tuned to a range where one bound state is tightly bound while a second is very close to the continuum of scattering states. Of course, for a real material reliable ab initio computations are a remarkable achievement. But the response in the rebuttal clearly underlines that the method has been introduced before and has been performed for over 100 magnetic systems.

We appreciate the referee's detailed consideration. While the theoretical existence of both Frenkel and Wannier-Mott exciton limits is established, we disagree that their co-existence in a single material is a trivial finding. From the perspective of Materials Science the discovery and characterization of two fundamentally different exciton species co-existing in a real material like CrSBr represents a significant and non-obvious experimental realization. Unlike theoretical models where parameters are freely tunable, this natural co-existence provides a unique platform for studying exciton physics which is additionally coupled to magnetic order. Importantly **the value of our work does not rest solely on the computational method itself but on the insight yielded by the method**, the prior use is meaningless in this context. Following the referee's logic, one could diminish countless physics studies because they rely on established frameworks like density functional theory or the fundamental laws of thermodynamics. **Our work's quality and novelty is based on its unprecedented success at linking the exciton character to the complex magneto-optical coupling in CrSBr, providing an essential and currently unavailable microscopic explanation.**

The response to the second Reviewer's question where the multiple peaks come from remains very vague:

of course, several bands and/or phonon assisted transitions may play a role. But please be definite: What does the theory tell us? Are there many X_A's and X_B or only one each?

The theory provides a clear, distinct prediction for the number of pure electronic exciton states in the antiferromagnetic AFM and FM phase, as shown in Figure 1(e). Theory predicts that in X_A energy range there are two excitons and around X_B one at AFM phase which evolves into a set of states at FM phase. The experimentally observed spectra are indeed more complex than these pure electronic states probably due to coupling of excitonic states with phonons and potential polaronic effects. **We stress that all theoretical predictions, experimental data, and discussions regarding these complexities and possible controversies are thoroughly presented and explained in the manuscript.**

Importantly the primary conclusion of our study—the distinct Frenkel-like and Wannier-Mott-like character of the two exciton families—remains robust. The existence of multiple transitions within each group does not weaken the fundamental distinction we have established between the two types of excitons and their drastically different magneto-optical responses.

The statements in the rebuttal on zero momentum of the exciton are strongly misleading. In order to achieve zero momentum from two constituents, they must have opposite momenta (or in case of particles and holes the same momenta) - the velocities are not important.

We agree with the referee. All our exciton plots clearly show the electron and hole states from same momenta take part in exciton formation.

At the outset, we would like to express our gratitude to the referees for their time and valuable feedback, which enabled us to significantly improve our work.

Reviewer #1 (Remarks to the Author):

I would like to express my sincere gratitude for the constructive discussion that transpired between the reviewers and the authors, centering on the novelty and significance of this scholarly endeavor. As a researcher who has worked on CrSBr, I appreciated the submitted work. Consequently, I reaffirm my original position and support the publication of this work.

We are grateful for such positive feedback on our efforts and clarification.

Reviewer #2 (Remarks to the Author):

As the authors provide no further experimental results to distinguish the in-plane feature vs the out-of-plane feature (The related experiments are mentioned in my first reply), I stay with my previous opinion.

A direct experimental mapping of the exciton wavefunction along both in-plane and out-of-plane directions would indeed be scientifically interesting. However, such measurements are extremely challenging, with no guarantee of success, and most importantly, are not essential to conclude about the Frenkel or Wannier nature of excitons. Conducting such experiments would significantly delay publication while providing only marginal additional insight into the physical picture that we propose.

We understand that the reviewer's concern relates to the earlier comment (first revision):

"8. The authors use Fig.2 to show that the in-plane radius of XB is a few times larger than XA. However, I think this is not adequate to explain large redshift in XB exciton as the AFM-FM transition induced redshift is related to the out-of-plane spread."

In response, we would like to reiterate what we believe is already clearly stated in the manuscript. The dominant origin of the AFM–FM excitonic shifts is the change of the bandgap when spin confinement is lifted, not the out-of-plane delocalisation of the exciton wavefunction. Any out-of-plane spreading has, at most, a secondary effect. Several points support this conclusion. Firstly, Interlayer delocalisation of XA is essentially absent in FM phase, yet both excitons exhibit shifts that follow the magnetic-order-induced changes in the band structure. Secondly, the shift of XB tracks the bandgap change almost one-to-one, which would not be expected if out-of-plane delocalisation were the primary mechanism.

We emphasise that once the spin constraints are relaxed, interlayer spin hopping becomes allowed, changing the bandgap; however, this change does not affect all excitons equally. XA weakly follows band edges as it is highly delocalized in k-space, while XB is more localised (in k-space) around band-edge-high-symmetry points and tracks the band gap positions. Crucially, these Frankel or Wannier characters of XA and XB can be directly probed by their in-plane extension, as is done in our work. Therefore, probing the *out-of-plane* wavefunction profile is not necessary to support our

interpretation of the excitonic response across the AFM–FM transition. This mechanism, where only a particular kind of exciton energy tracks the band edge due to changes in spin-dependent hopping, is subtle and, in our view, has not been clearly articulated in prior works. We respectfully suggest that this nuance may also have been underappreciated in the referee’s interpretation

In the end, we want to stress that although we cannot experimentally quantify the out-of-plane extension of the exciton wavefunction in the FM phase, we do observe indirect experimental evidence that XB is more out-of-plane extended than XA. This is reflected in its enhanced coupling to out-of-plane phonon modes, a phenomenon we have explored through both experimental observations and theoretical modelling. While our high-field measurements do not directly map the out-of-plane spatial extension of the XB exciton, we believe the observed energy shifts and phonon coupling provide compelling indirect evidence for this evolution. We therefore feel that this limitation should not be viewed as a shortcoming of our work.

Response to Reviewers

Reviewer #1 (Remarks to the Author):

I found that Reviewer #2 had a different opinion regarding the evaluation of the manuscript. Based on my own judgement, however, I stand by my original position and support the publication of this work.

We thank Reviewer #1 for maintaining their original opinion and for their continued support of this manuscript.